# First insights into Northern Africa high-altitude background aerosol chemical composition and source influences

Nabil Deabji[1,2], Khanneh Wadinga Fomba[1], Souad El Hajjaji[2], Abdelwahid Mellouki[3], Laurent Poulain[1], Sebastian Zeppenfeld[1], and Hartmut Herrmann[1]

[1]Leibniz Institute for Tropospheric Research (TROPOS), Atmospheric Chemistry Department (ACD), Permoserstraße 15, 04318, Leipzig, Germany
[2]LS3MN3E-CERNE2D, Faculty of Science, Mohammed V University in Rabat, 4 Avenue Ibn Battouta, B.P. 1040, 10100 Rabat, Morocco
[3]Institut de Combustion Aérothermique Réactivité et Environnement/OSUC-CNRS, 1C Avenue de la Recherche
Scientifique, 45071 Orléans Cedex 2, France

*Correspondence to:* Hartmut Herrmann (herrmann@tropos.de)

**Abstract.** Field measurements were conducted to determine aerosol chemical composition in a newly established remote high-altitude site in North Africa, at the Atlas Mohammed V atmospheric observatory (AMV) located in
the Middle-Atlas Mountains. The main objectives of the present work are to investigate the variations in the aerosol composition, and better assess global and regional changes in atmospheric composition in North Africa. A total of 200 Particulate matter ($PM_{10}$) filter samples were collected in at the site using a high-volume (HV) collector in a 12h sampling interval from August to December 2017. The chemical composition of the samples was analyzed for trace metals, water soluble ions, and organic carbon (OC/EC), aliphatic hydrocarbons, and
polycyclic aromatic hydrocarbon (PAHs) contents.

The results indicate that high-altitudes aerosol composition is influenced by both regional as well as trans-regional transport of emissions. However, local sources play an important role, especially during low wind speed periods, as observed for November and December. During background conditions characterized by low wind speeds (Av. 3 m s$^{-1}$) and mass concentrations in the range from 9.8 to 12 µg m$^{-3}$. The chemical composition is found to be
dominated by inorganic elements, mainly suspended dust (61%) and ionic species (7%), followed by organic matter (7%), water content (12%), and unidentified mass (11%). Despite the proximity of the site to the Sahara Desert, its influence on the atmospheric composition at this high-altitude site was mainly seasonal and accounted for only 22% of the sampling duration. Biogenic organics contributed up to 7% of the organic matter with high contributions from compounds such as heneicosane, hentriacontane, and nonacosane. The AMV site is dominated
by four main air mass inflow, which often leads to different aerosol chemical compositions. Mineral dust influenced was seasonal and ranged between 21 and 74% of the PM mass with peaks observed during the summer and was accompanied by high concentrations of $SO_4^{2-}$ of up to 3.0 µg m$^{-3}$. During winter, $PM_{10}$ concentrations are low (< 30 µg m$^{-3}$), the influence of the desert is weaker, and the marine air masses (64%) are more dominant with a mixture of sea salt and polluted aerosol from the coastal regions (Rabat and Casablanca). During the
daytime, mineral dust contribution to PM increased by about 42% because of road dust resuspension. In contrast, during night-time, an increase in the concentrations of alkanes, PAHs, alkane-2-ones, and anthropogenic metals such as Pb, Ni, and Cu was found due to variations in the boundary layer height. The results provide the first

detailed seasonal and diurnal variation of the aerosol chemical composition, valuable for long-term assessment of climate and regional influence of air pollution in North Africa.

## 1    Introduction

Aerosols are important constituents of the atmosphere due to their role in controlling climate processes and their impact on air quality, the environment, and ecosystems. They can have adverse effects on human health and have been associated with respiratory disorders, strokes, pulmonary and cardiovascular diseases (Du et al., 2016; Pope et al., 2018; Song et al., 2014). Aerosol particles can serve as cloud condensation nuclei and as substrates for heterogeneous reactions (Leng et al., 2014). Their chemical composition affects aerosol-cloud interaction and may exert a warming or a cooling influence on the atmosphere due to direct and radiative forcing (King et al., 2003; Satheesh and Krishna Moorthy, 2005). Therefore, the study of aerosol chemical proprieties is essential for a better understanding of atmospheric processes.

Atmospheric aerosol particle composition depends on local and regional emission sources as well as transboundary pollution. The particles are emitted directly into the atmosphere from natural sources such as sea salt or mineral dust and anthropogenic activities such as industrial or traffic emissions, constituting primary emissions. They can also be formed in the atmosphere through gas-to-particle conversion or particle-phase reactions, constituting secondary aerosols (Carter et al., 2005). After emission, these particles are exposed to changing humidity, temperature, pressure, and solar radiation in the atmosphere that alters their properties through different aging and oxidative processes during atmospheric transport. Consequently, high-altitude sites provide the required infrastructure for investigating and characterizing the possible atmospheric aerosol interactions associated with the particles.

High-altitude sites in remote regions are less affected by direct local anthropogenic emissions. Their high altitude allows the study of aerosol particles in the free atmosphere and provides a good impression of aerosol background concentrations. The topography, meteorological conditions, and changing boundary layer heights provide various pathways for aerosol interactions, which could influence inversion processes and enhance biogenic particle formation. Such sites are hence unique for monitoring the temporal variation in the aerosol chemical compositions over longer periods and provide a better understanding of various factors, such as meteorology, climate, and environmental changes that may in the long-term affect the local and regional air composition (Okamoto and Tanimoto, 2016).

There is increasing interest in atmospheric aerosol studies at high-altitude sites. Studies have investigated the aerosol chemical composition in mountainous regions, highlighting the influence of mountain valleys, night-time mountain breeze, and topography in dispersing polluted air masses to the free troposphere (Zhang et al., 2009; Alastuey et al., 2005; Buchunde et al., 2019; Leena et al., 2017; Glasius et al., 2018; Lugauer et al., 1998; Mukherjee et al., 2020). Furthermore, other studies at the Northeastern Himalayas, India (Chatterjee et al., 2010), the Bachelor Observatory Mountain in Oregon, USA, (Ambrose et al., 2011), a Mountain site at Lulang on the southeast Tibetan Plateau, China (Zhao et al., 2013) have reported the importance of the aerosol chemical composition at mountain sites in the identification of potential source regions of anthropogenic pollutants and their mechanism of transport. Some observations and models have elaborated the emissions of some trace gases such as CO and $O_3$ from the boundary layer into the free troposphere by convective, frontal, and orographic lifting

at mountain sites (Bey et al., 2001; Ding et al., 2015; Liang et al., 2004; Weiss-Penzias et al., 2006). Nevertheless, the effects of these mechanisms have rarely been studied at the aerosol chemical composition scale. Despite the increasing interest in high-altitude aerosol research, most studies have reported measurements of tracer gases over long-term periods, but limited studies have addressed the interaction between natural emissions such as mineral dust, biogenic compounds, and anthropogenic emissions in the free troposphere (Fiore et al. 2009; Jonson et al. 2010; Logan et al. 2012; Gilge et al. 2010; Kumar et al. 2013).

Moreover, such studies have been reported mostly in central Europe, Asia, and North America (Okamoto and Tanimoto, 2016). A few attempts in Africa have been made to investigate the microphysical and optical properties of mineral dust transport in North-Africa (Kandler et al. 2009; Müller et al. 2012; Ryder et al. 2011; Schladitz et al. 2009; Veselovskii et al. 2016). Other studies have focused on polluted regions (Benchrif et al., 2018; Inchaouh, 2017; Tahri et al., 2013), such that information about the chemical composition of particulate matter at high-altitudes is limited. Likewise, background aerosol information, essential in assessing long-term regional changes in atmospheric composition in this region remains scarce and difficult to assess due to lack of the necessary infrastructure. This poor state of knowledge limits transregional investigation on the effect of different sources and source regions on the chemical composition of aerosol particles over sensitive regions such as the Atlas Mountains in North Africa.

The Middle-Atlas region located in the North of Morocco is typically considered as an area with high rainfall. Still, according to the report from the Moroccan ministry of environment, the annual average rainfall decreased by about 100 mm, with an increase in temperature by about 1.5 °C (Royaume du Maroc, 2009) within the past 50 years. These are indicators of the sensitive nature of the Middle-Atlas region to a changing climate. The Ifrane national park, which is located in the Middle-Atlas, suffers from intense pressure due to forest degradation, and overgrazing resulting in significant climatic consequences (Campbell et al., 2017). At the same time, it is classified as a site of biological and ecological interest. Recently, soil erosion was particularly intense in some clay-dominated valleys (Mounir et al., 2019). This change can lead to several consequences such as an increase in aridity, reduction of precipitation, change in primary emissions, and atmospheric composition. Thus, the observations from this region could provide new knowledge into atmospheric composition changes over time, related to different climatic and anthropogenic dynamics. The evaluation of local regional, trans-regional, and climate change effects can help to assess air quality and climate-relevant mitigation strategies.

The aim of this study was, therefore, to i) quantify and characterize the variability of $PM_{10}$ mass concentration in the high-altitudes of the Middle-Atlas region, ii) determine their chemical composition, iii) identify the possible sources of the aerosol particles, and iv) evaluate the relative contributions of the source regions to the observed concentrations. Within the present study, chemical composition at the high altitude AMV observatory is presented. Chemical components such as trace metals, OC, EC, ionic, and organics species were investigated, and meteorological and back trajectory analysis was performed. Moreover, the influence of dust on the chemical composition and the day and night variation of the $PM_{10}$ concentration were investigated.

## 2    Experimental

### 2.1    Site description and particles sampling

The Atlas Mohammed V (AMV) atmospheric research station situated in a strategic location in the Middle Atlas was founded in 2017. It is operated by the Centre National de la Recherche Scientifique (CNRS-ICARE, Orléans-France), the Mohammed V University (Rabat-Morocco), and the Leibniz Institute for Tropospheric Research (TROPOS, Leipzig-Germany). The observatory is located at Michlifen in the Middle Atlas region at an altitude of 2100 m a.s.l in a remote hilly site (33°24'22.2''N 5°06'12.0''W). On one side, are the plains and plateaus of central Atlantic Morocco, and on the other side, the arid areas. It is about 300 km north of the Sahara desert, about 230 km east of the Atlantic Ocean and the populated and industrial regions of Casablanca and Rabat, and about 340 km south of the Mediterranean Sea. The orientation of the Middle Atlas Mountains diagonally extends from southwest to northeast over a distance of 450 km. The AMV station is surrounded by cedar forests and pastureland that come to life in spring and summer, which is a sharp contrast to the hot, dry climate surrounding it. The nearest urban towns are Ifrane, and Azrou, which are about 22 km away, and Fes city located 82 km north of the AMV station. Due to its remote location, the influence of anthropogenic emissions from the Ifrane area is low.

Aerosol particles were sampled using a $PM_{10}$ high volume Digitel (DHA-80, Switzerland) with a flow rate of 500 l/min on quartz fiber filters (Munktell, MK 360). The collection period was from August to December 2017, during which 200 filters have been collected in a day and night-time (12 h) sampling routine. At the end of the sampling, the collected filters were placed in a refrigerator at a temperature of 5°C and subsequently frozen at - 20°C. After storage, the filters were transported at -20°C to TROPOS (Leipzig-Germany) in aluminum cans for chemical analysis. In September due to power failure and instrument outage fewer samples were collected (n=20) Due to the lower number of samples, the missing days of sampling were replaced by the average concentration of the collected samples in September.

### 2.2    Local meteorology and station characteristics

Meteorological data were collected from summer 2017 to spring 2018 for major parameters such as temperature, relative humidity, wind speed and direction, atmospheric pressure, visibility, and precipitation at a sampling rate of 1 min using an automated weather station (Bresser AWS, Germany).  As shown in Table 1, in summer, temperatures are moderate or warm during the day and cool at night. Winter is much colder, and the daily temperature amplitudes are lower because the valleys only receive the sun's rays in the middle of the day.  In winter, the station remains entirely in the shade for several weeks. Thermal contrasts between slopes are important when the topographic terrain is oriented east-west. Thermal breezes are common during high-pressure weather in the valleys. The temperature varies seasonally, especially during the transition from summer to winter, with maximum and minimum values of 26°C and -1°C, respectively. The annual average temperature is approximately 14°C, with a sharp decrease during the night.

In contrast, the visibility varies slightly, with intense UV radiation during summer when the sky is often clear. Fog occurrence is high during autumn and winter. The wind comes from all directions, but it is dominated by air mass from the west, as shown in Fig. 2. The average wind speed at AMV was about 5.8 m s$^{-1}$ but reached a

maximum of 19.7 m s$^{-1}$ due to turbulence in the mountain region, especially during winter. Over the summer, the minimum wind speed was about 1.6 m s$^{-1}$, and the relative humidity (RH) was low. In autumn, RH was on average about 36% and reached up to 97% under the influence of marine air mass in winter. The Middle-Atlas region is considered to be a very humid and with temperate climate. Indeed, the water balance required for plants is mainly positive in winter, about 141 mm, while annual precipitation is 300 mm. Rainfall occurs mainly during winter,

with heavy thunderstorms and a lot of snow coverage. The northern part of the Middle Atlas Mountains is the wettest region in Morocco after the Rif mountain regions, according to Nourelbait et al. (2016). Precipitation increases in frequency and intensity during the winter. Indeed, the mountain imposes an ascent of air masses, which results in cooling, the formation of clouds, and the condensation of water vapor. The proportion of snowfall also increases rapidly because of the altitude, especially in winter. On the other hand, no difference was observed

for the visibility during the seasons which shows an average value of 10 km.

There is a wide variation in local wind distribution over seasons where the wind comes from all directions, except the north. During summer, southeast winds have a higher frequency but a lower average speed of about 5.8 m s$^{-1}$. Similarly, southeast winds are dominant with a slight decrease in wind frequency during the fall while western

winds are dominant by up to 30% during winter and spring. During this period, there is a strong occurrence of westerly winds which are often characterized by high wind speeds (stiff breeze) of up to 20 m s$^{-1}$. To conclude, the wind frequently comes from the west and southwest during the winter months, in contrast to the dominant south winds in summer.

### 2.3    Aerosol particle chemical analysis

*Particle mass*

The collected filters were weighed on a microbalance with an accuracy of 10 µg (Mod. AT261 Delta Range, Mettler) after being stabilized for 72 hours at constant temperature (20±1 °C) and humidity (50±5%), before and after sampling. The difference between the weights was determined and divided by the total sampling volume to obtain the mass concentrations. After the determination of the particulate matter concentration, both organic and

inorganic analyses were carried out at the TROPOS laboratories.

*Carbon compounds*

Organic and elemental carbon were analyzed by a thermo-optical method (Sunset Laboratory Inc. U.S.A) at a maximum Temperature of 850°C with the normalized temperature program EUSAAR2 (EUropean Supersites for

Atmospheric Aerosol Research), as described in the literature (Cavalli et al., 2010; Yttri et al., 2019). The method is in line with the standard proposed by the European networks (ACTRIS, EMEP). Samples were thermally desorbed from the filter medium under an inert He-atmosphere followed by an oxidizing $O_2$/He-atmosphere using carefully controlled heating ramps. A flame ionization detector is used to quantify methane after catalytic methanation of $CO_2$. First, the sample is heated up to about 870°C in an inert atmosphere of pure helium. These

conditions allow the organic carbon to volatilize and to be fed into the second furnace filled with $Mn_2O$ (oxidation catalyst), where it is quantitatively oxidized into $CO_2$. As a second step, the sample is placed in an oxidizing atmosphere (helium/oxygen), leading to the oxidation and volatilization of the refractory elemental carbon remaining on the filter (van Pinxteren et al., 2015). Charring processes lead to the overestimation of EC and an

underestimation of OC, resulting in lower OC/EC ratios. Therefore, an optical correction was applied for the charring process. The optical correction of charring for pyrolytic carbon is obtained by measuring the transmission of the sample with a laser (wavelength 678 nm). The detection limit for OC/EC measurement was 0.2 $\mu g/cm^2$. Organic matter (OM) was estimated based on the $f_{OM/OC}$ conversion factor, according to Turpin and Lim (2001). The total carbon TC was considered as the sum of organic carbon and elemental carbon (TC=OC+EC). Organic matter was about twice the organic carbon (OM = 2.1 × OC). Since the conversion factor depends on the specific proportion of each site, the factor $f_{OM/OC}$ = 2.1 is suggested because it takes into account aged aerosols (Turpin and Lim, 2001). The primary organic carbon (POC) fraction in the $PM_{10}$ was estimated using EC as a tracer, by taking the minimum OC/EC ratio for the entire study period and multiplying it to the EC content, as in the following equation POC = $(OC/EC)_{min}$ × EC. Consequently, the secondary organic carbon (SOC) contribution to the total OC can be estimated as the difference between the total organic carbon and total primary organic carbon concentrations (SOC = OC-POC).

Organic compounds such as n-alkanes oxygenated polycyclic aromatic hydrocarbons (oxy-PAHs) and n-alkane-2-ones were detected using a Curie-point pyrolyzer (JPS-350, JAI Inc., Japan) coupled with a GC-MS system (6890 N GC, 5973 inert MSD, Agilent Technologies, CA, USA) as described by Neusüss et al., (2000). Saccharidic compounds such as mannitol, glucose, levoglucosan, and arabitol were determined using high-performance anion-exchange chromatography with pulsed amperometric detection (HPAEC-PAD) as described by Iinuma et al., 2009.

*Trace metals*

Trace metals were determined using the Total Reflection X-Ray Fluorescence technique (TXRF), whereby 3 spots of 8 mm in diameter each were digested in 1.125 mL $HNO_3$ and 0.375 HCl using the Mars 6 (CEM, Germany) microwave. 50 µL of the digested solution was deposited on previously siliconized quartz carriers, and 10 ng of Galium was added onto the sample as an internal standard. The samples were subsequently measured using an S2-PICOFOX (Bruker AXS Microanalysis GmbH, Germany) instrument. Further details of the technique and measurement procedure have been reported elsewhere (Fomba et al., 2020).

*Water-soluble ions*

The major ionic constituents were analyzed using a standard ion chromatography technique (ICS3000, Dionex, USA) equipped with automatic eluent generation (KOH for anions and methanesulfonic acid (MSA) for cations) and a micro membrane removal unit. Ion analysis was performed for $Na^+$, $NH_4^+$, $K^+$, $Mg^{2+}$, $Ca^{2+}$ cations, and $Cl^-$ and $Br^-$, $NO^-$, $SO_4^{2-}$ and $C_2O_4^{2-}$ anions. For these analyses, 3 spots of 2 cm each in diameter were extracted from the filter in deionized water via shaking for 2 h. The extract was filtered through a 0.45 µm unidirectional syringe filter to remove insoluble matter, and the filtrate was analyzed. The blank field filters were analyzed using similar procedures and were subtracted from the sample concentrations following the methodology described by Iinuma et al. (2009).

*Estimation of Sea salt components*

Sea salt concentrations were calculated by adding chloride to sodium, and the sea salt (ss) contributions of potassium (ss-$K^+$), calcium (ss-$Ca^{2+}$), magnesium (ss-$Mg^{2+}$), and sulfate (ss-$SO_4^{2-}$), , have been estimated as 0.03, 0.5, 0.12, and 0.25 fractions of the measured $Na^+$, respectively (Marenco et al., 2006). However, the estimation of sea salt assumes that all Na and Cl are coming from marine contribution without taking into consideration other potential sources. The estimation of non-sea salt sulfate (nss-$SO_4^{2-}$) was estimated by subtracting the contribution of ss- $SO_4^{2-}$ from the total $SO_4^{2-}$ mass concentration (Amodio et al., 2014). The water content of the samples was estimated according to the E-AIM (Extended Aerosol Inorganics Model) III of Clegg et al. (1998).

## 2.4    Determination of mineral dust

Mineral dust (MD) is a significant contributor to atmospheric particulate matter especially in North-Africa, where the average percentage can vary from 7% to 62% (Gherboudj et al., 2017). The high spatiotemporal variability of the dust emission can sometimes be difficult to quantify correctly and may lead to some uncertainty. Therefore,the estimation of MD can be subjective because several estimation methods are available in the literature. The most common methods were applied to the samples collected at AMV station. Within the present study, the aim was first to evaluate and then select an appropriate method for the interpretation of the results. Four methods were highlighted that are representative of those used in the literature. The first method implemented by Fomba et al., (2014) consists of subtracting the total $PM_{10}$ mass concentration from the analyzed mass of the other elements, representing the upper limit of the possible MD concentration in the samples. This method is an interesting approach especially when the elemental analysis is not available. However, to improve  the MD quantification using available MD-related elements, applying a stoichiometric equation reduces the uncertainty of the values. Using this method, an average MD concentration of about 24.5 µg $m^{-3}$ was obtained. In contrast, methods 2 and 3 use different approaches. They estimate MD base on given stoichiometry and apply different elemental concentrations such as Al or Ca, Fe, Ti to estimate the MD load. Method 2 uses the factor (1.16) to compensate for the exclusion of MgO, $Na_2O$, $K_2O$, and $H_2O$ from the crustal mass calculation, as shown in Table 2 (Maenhaut et al., 2005). Whereas method 3 considers carbonate such as calcite, dolomite, and other oxides such as $TiO_2$, $Fe_2O_3$, and $MnO_2$ (Minguillón et al., 2007; Nerriere et al., 2007). As a result, the average MD concentration using methods 2 and 3 were similar, about 19.9 and 18.9 µg $m^{-3}$, respectively. Method 4 takes into account that sea salt significantly affects these concentrations, and the non-sea salt (nss) content of the elements, such as nss-$Ca^{2+}$ or nss-$Mg^{2+}$ are used to replace the total Ca and Mg concentrations. This is more accurate in a sea-salt-dominated environment but could underestimate the calcium contribution as sodium also has a crustal origin (Cesari et al., 2012; Perrino et al., 2014). The average MD concentration from this method was about 15.5 µg $m^{-3}$.

In conclusion, differences of up to 37% were obtained between these methods. The obtained MD concentrations were high for method 1 and lowest for method 4 and similar between methods 3 and 4, indicating their robustness. Method 3 does not consider quantifying silicates $SiO_2$ and aluminosilicate $Al_2 SiO_2$, a major component in natural mineral dust. However, method 2 allows to take into account the overall mineral composition and therefore was applied in this study. Due to limitations in quantifying Si from quartz fiber, MD was finally estimated by replacing "Si" with "Ca" base on the established soil stoichiometric ratio of the average upper continental crust (Si=10.3 Ca), according to Wedepohl (1995) in the equation used in Method 2. The final equation used for mineral dust estimation is given as follows:

$$Mineral\ dust = 1.16(1.90\ Al + 23.3\ Ca + 2.09\ Fe + 1.67\ Ti),\qquad\qquad (1)$$

### 2.5    Back trajectory analysis

The origin of the air masses reaching the station, 96h back trajectories were estimated using the NOAA HYSPLIT (Hybrid       Single-Particle       Lagrangian       Trajectory,       HYSPLIT       4)       model (http://www.ready.noaa.gov/ready/hysplit4.html; Draxler and Hess, 2004), using the 1° resolution Global Data Assimilation System (GDAS) input data. Due to the resolution of the input data, the exact altitude of the mountain is not properly represented and according to the HYSPLIT model at the AMV site, the terrain height is at 1000 m

only. Therefore, trajectories were calculated every hour for the altitude of 1000 m above model ground level. The profile of the air mass altitude during transport was applied for the interpretation of the data for each air mass.

To group the back trajectories into distinct transport patterns, a manual classification approach was used. This method consists of grouping 12 trajectories with the time interval between adjacent nodes of 1 h and calculated for 96 h, and attributing them to a specific air mass category. The assignment of the trajectories was based on

their crossing over given latitude-longitude grids attributed to given geographical sectors. To assign a sample to a specific air mass category, 60% of the trajectories must have a similar profile. Samples (n=15 samples) with trajectories from mixed origins (e.g., marine air mass over Europe and the desert) were excluded from the classification of the air masses. In total, the air masses of 175 samples could be grouped into four distinct categories which were Background Air mass (BAM), Atlantic Coast Europe (ACE), Mediterranean Coast Europe

(MCE), Saharan Dust (SD).

### 3    Results and discussion

### 3.1    Variation of $PM_{10}$ mass

The $PM_{10}$ mass concentration time series at the AMV station varied from 9.5 µg m$^{-3}$ to 145.6 µg m$^{-3}$ with averaged of 29.2 ± 17.3 µg m$^{-3}$. The $PM_{10}$ mass shows a strong seasonal variation during the five months of measurement

from August to December 2017, as illustrated in Fig. 3a.  The highest monthly concentration was observed in August (49.9 ± 25.9 µg m$^{-3}$) and continuously decreased until December (15.9 ± 5.6 µg m$^{-3}$), as shown in Fig. 3b. The observed temporal variation in the concentration is most likely related to factors such as meteorological conditions and the air mass arriving at the station. To understand the seasonal variation of the particulate matter load, $PM_{10}$ mass was combined with wind speed and wind direction to create the polar plots as presented in Fig.

3c, which illustrates the variation of $PM_{10}$ concentrations as a function of wind speed and direction.

During August, the high $PM_{10}$ concentrations were mostly related to high wind speeds from the southeast. For example, $PM_{10}$ mass concentration often exceeded 50 µg m$^{-3}$ and sometimes even reached up to 145 µg m$^{-3}$ during August, when the wind speed was stronger than 9 m s$^{-1}$. The high $PM_{10}$ concentration recorded was due to the

influence of Saharan dust events during periods of air mass influence from the southern sector located in the southeast of the AMV station. The Middle Atlas region is marked by particular meteorological conditions during the summer with low humidity and often low precipitation (avg. 37 mm), as shown in Table 1. These hot and arid conditions are known to favor the transport of dust particles from the Saharan desert to the Atlas Mountains

(Rodríguez et al., 2011). Furthermore, high concentrations of up to 40-50 µg m$^{-3}$ were observed as well with westerly winds, especially during northwest and southwest winds. High PM$_{10}$ concentrations were observed during strong westerly winds of up to > 7 m s$^{-1}$. The back trajectory analysis suggests that the high concentrations during this period were most likely associated with the long-range transport of aerosol particles from the western coast of the Iberian Peninsula. In contrast to the summer period, PM$_{10}$ mass concentrations were lower during the fall, despite some temporal peaks. The PM$_{10}$ concentrations were generally lower in September (24.2 ± 5.1 µg m$^{-3}$) and October (30.5 ± 10.8 µg m$^{-3}$). During this period, wind originated from the northeast suggesting the influence of air mass transport from the Mediterranean Sea coast. A sharp fall in PM concentrations was noticed in November (22.8 ± 7.9 µg m$^{-3}$) and December (15.9 ± 5.6 µg m$^{-3}$). Overall, PM$_{10}$ concentration decreased from the summer to winter by 32%. This trend is most likely due to the increased amount of precipitation (peaks of 852mm) during fall and winter, which can lead to the wash-out effect of aerosol and its components (Holst et al., 2008).

To establish a reference baseline and evaluate the background conditions at the site, the lower 5$^{th}$ percentile of the PM$_{10}$ concentrations (PM$_{10}$ < 12 µg m$^{-3}$) was found to be representative of remote background aerosol conditions. The PM$_{10}$ frequency and probability density function as shown in Fig. S3 confirmed this observation. The samples within this PM concentration range had similar air mass trajectories and typical meteorological conditions with low wind speeds < 3 m s$^{-1}$. The air masses typically traveled in the free troposphere at about 1000 m above sea level, crossing the North-Atlantic Ocean before arriving at the site within the past 96 h. These conditions were, however, not free from local and regional pollution from point sources such as dust resuspension from the cars assessing the site.

Consequently, the average background PM$_{10}$ mass concentration at the AMV was 10.9 µg m$^{-3}$, which was found to be stable and representative of periods of little external influence. In comparison, Benchrif et al. (2018) reported background PM$_{10}$ values for Northern Morocco with an average of 12.2 µg m$^{-3}$, which is very similar to the concentrations determined in this study, 10.9 µg m$^{-3}$.

The mean concentration recorded at AMV from this study (29.2 ± 17.3 µg m$^{-3}$) agreed well with the PM$_{10}$ concentration of other remote high-altitude sites, such as Darjeeling in Northeastern Himalayas (29 µg m$^{-3}$; Chatterjee et al. 2010), Lhasa in Tibet (37 µg m$^{-3}$; Wang et al., 2015), and Mahabaleshwar in India (37 µg m$^{-3}$; Leena et al., 2017) as presented in Table 3. Other high-altitude stations, such as Izaña, in Canary Islands (46 µg m$^{-3}$) showed much higher PM$_{10}$, most likely due to the exposure to strong Saharan dust events (García et al., 2017). In contrast, the PM$_{10}$ concentrations at AMV were considerably higher than the PM$_{10}$ levels recorded in European and Asian high-altitude sites. For example, the average PM$_{10}$ mean value recorded in this study was about twice that of Mount Cimone, Italy (16 µg m$^{-3}$; Marenco et al., 2006) and factor 6 greater than the PM$_{10}$ in Everest Mountain (Decesari et al., 2010) and in Puy de Dôme, France (6 µg m$^{-3}$; Bourcier et al. 2012), and was approximately 10 times greater than the average level in Jungfraujoch (3 µg m$^{-3}$; Cozic et al., 2008). Other Moroccan sites, such as Marrakech, Meknes, and Agadir, which are exposed to strong urban emissions, usually show PM$_{10}$ concentrations between 50 and 110 µg m$^{-3}$, which are much higher than the concentration found at AMV in this study (Inchaouh, 2017; Tahri et al., 2013, 2017). These results highlight a better air quality at AMV

in comparison to many sites and indicate that the station can serve as a good remote reference station for defining background concentrations in Morocco and possibly the whole of North Africa.

## 3.2    Air mass origins

The calculation of back trajectories using the HYSPLIT model allowed the identification of several remote sources of PM10 at the station. Four main air masses categories were identified as shown in Fig. 4. i) Air masses that spent the last 96h over the Atlantic Ocean at high-altitude (1000 m.asl), representative of typical background air mass (BAM) conditions, which influenced about 5.3% of all samples; ii) Air masses originating from the Atlantic and crossing over the Coast of Europe (ACE), especially Spain and over Moroccan industrial cities located at the North Atlantic coast, influencing about accounts 26.8% of all samples; iii) Air masses from Europe crossing over the Mediterranean Coast and Europe (MCE) as well as over North Moroccan cities, as shown in Fig. 4c and influencing about 37.4% of all samples and iv) Air masses originating from Southern and/or Eastern Sahara crossing the desert (SD), at different altitudes before arriving at the AMV and influencing about 22.6% of all samples. The back trajectories represent mixing scenarios (7.9% of all samples) were not assigned to any of the four major classes as mentioned above.

## 3.3    Characterization of aerosol chemical composition

The statistics of the measured PM10 chemical components during the four air mass categories are shown in Table 4, including the average concentrations and their variations (Std). Likewise, Fig. 6 shows the time series of the investigated chemical species within the sampling period colored with periods of the different air mass influence.

### 3.3.1 Mineral dust

During the 5 months of PM collection at the AMV site, the average mineral dust concentration was about $17.7 \pm 7.4$ µg m$^{-3}$ and varied strongly between 0.05 µg m$^{-3}$ and 107 µg m$^{-3}$. The highest mean concentrations were observed in August (39 µg m$^{-3}$) and the lowest in December (3.7 µg m$^{-3}$). Low concentrations were observed during days with low wind speeds (< 2 m s$^{-1}$), low Saharan dust air mass inflow, and after precipitation events, which typically occurred in the fall and winter. The influence of the Saharan dust on the Middle Atlas region remains relatively dependent on meteorological conditions. Firstly, the direction and speed of the wind, as the typical Saharan dust events were observed during high wind speed periods from south and southeast. Secondly, their progression depends mainly on favorable weather conditions for transport, the difference in temperature between day and night, humidity, and especially the scarcity of rainfall. Thirdly, the High-Atlas mountains situated at 4000 m of altitude act as a barrier to Saharan dust transport which forces the winds to deviate from their path. All these factors influence the transport of large particles from the Sahara to the Middle-Atlas during the different seasons.

Nevertheless, even during days of low wind speed, mineral dust still dominated the aerosol composition and contributed up to 51% of the total mass, as shown in Fig. S4, for typical background conditions chemical composition. The highest aerosol mass was observed during days of Saharan dust events from the 10$^{th}$ to the 13$^{th}$ of August when air mass crossed the Sahara (SD) before arriving at the AMV site. The duration of Saharan dust events varied from 1 to 3 days, with the longest event (also supported by back trajectory analysis) observed during

August. Most dust events occurred during the summer, as indicated by the strong increase of the typical crustal elements (Guimot et al. 2007; Arimoto et al., 2006) such as nss-$Ca^{2+}$, Fe, and (Fig. 6). Mineral dust was found to be more than 7 times higher ($37.9 \pm 25.3$ µg m$^{-3}$) during dust events (SD), in comparison to remote background conditions (BAM) with an average concentration of $5.5 \pm 3.5$ µg m$^{-3}$, as observed in Table 4. Other less intense Saharan dust storms occurred during the summer season between the 21$^{st}$ and 24$^{th}$ of August with a similar high dust concentration that was 5 times higher than background dust concentrations. The presence of mineral dust is relatively low but still significant for air masses other than SD, such as during the ACE ($13.3 \pm 5.2$ µg m$^{-3}$) and MCE ($19.9 \pm 11.9$ µg m$^{-3}$) air mass influences, as shown in Table 4. This suggests that the air masses were very often loaded with mineral dust originating from regional sources in the Middle Atlas. The Fe/Ca ratio was used to distinguish between different mineral dust sources. Fe/Ca ratios close to 0.4 indicate dust from the south, while Fe/Ca ratios greater than 1 indicate dust from east. The Fe/Ca ratio is about 0.5 during BAM conditions, indicating local sources emitted by road dust or resuspension of agricultural activities.

Mineral dust can be transported over long distances particularly from the North African source region to the Mediterranean basin and Europe (Schepanski et al., 2016). For instance, the high-altitude site in Mt. Cimone, Italy recorded several days with African dust transport which influenced the chemical composition (Marenco et al., 2006). However, the Saharan dust concentration at AMV (17.7 µg m$^{-3}$) is approximately 4 times higher than at Mt. Cimone (4 µg m$^{-3}$). Nevertheless, the average concentrations of elements such as Al, Fe, Ti, and Mn, are comparable with the values reported in Mt. Cimone, Italy (Marenco et al., 2006), Table 5. However, the calcium, concentration at the AMV ($0.65 \pm 0.58$ ng m$^{-3}$), was 2 times higher than the concentration recorded in Mt. Cimone. Furthermore, the calcium concentration was 5 times higher than the concentration recorded at other high-altitude station such as Mt. Himalaya and Mt. Everest (Chatterjee et al. 2010; d; Decesari et al., 2010). This suggests that the AMV experiences higher amounts of calcium-rich dust in comparison to other sites. Some studies have reported the high content of calcite in the soils of Northern Morocco 1.07 ng m$^{-3}$ which confirms the predominance of calcium-rich in the Atlas regions (Desboeufs and Cautenet, 2005; Kandler et al., 2009; Benchrif et al., 2018).

**3.3.2 OC and EC**

Organic carbon (OC) and elemental carbon (EC) showed strong variation and distinct differences with an average of $1.1 \pm 0.8$ µg m$^{-3}$ and $0.2 \pm 0.1$ µg m$^{-3}$, respectively. The OC has both primary and secondary origin, and can be formed from primarily emitted substances through condensation or chemical reactions among them (Sarkar et al., 2019). The OC concentration reached a maximum 4.5 µg m$^{-3}$ during summer, whereas the lowest concentration was observed during winter at about 0.03 µg m$^{-3}$. The average concentration of OC progressively decreased from summer ($2.1 \pm 0.8$ µg m$^{-3}$) to winter ($0.3 \pm 0.2$ µg m$^{-3}$). The abundant contribution of organic matter in summer can be due to high biogenic emissions in the Middle Atlas. A slight increase in OC was also observed during dust events, as shown in Fig. 5, which suggests that the dust deposited at AMV also contained biogenic material from the surroundings of the Middle-Atlas region. The average concentration of POC was found to be $0.2 \pm 0.3$ µg m$^{-3}$ (26% of OC concentration), whereas SOC was estimated as $0.8 \pm 0.7$ µg m$^{-3}$ (74% of OC).

The EC concentration showed little temporal variation except for a few pollution episodes during which peaks in the EC concentrations could be observed (Fig. 5). The sudden increase in EC observed during summer (0.6 µg m$^{-}$

$^3$) and autumn (0.7 µg m$^{-3}$) were characterized by two different sources. Firstly, from Europe through the Atlantic Coast during ACE air mass influence in summer and secondly, from nearby urban regions especially in the evenings when the temperature is low (with an average of 5°C) and high values of anthropogenic metals, such as Pb, Cu, and Ni, observed. The wind direction and back trajectory analysis indicate that the most likely sources of pollution are the urban cities of Fes and Meknes, located about 85 and 50 km in the North from the station.

However, the winter period was marked by low EC concentrations, with an average of 0.1 ± 0.06 µg m$^{-3}$. The elemental carbon concentration EC concentrations (0.2 µg m$^{-3}$) during Saharan dust air masses were low indicating that the influence of urban pollution during dust events was low. Few studies observed that the transport of North African dust was often loaded with pollutants (Gangoiti et al., 2006; Kalderon-Asael et al., 2009; Astitha et al., 2010). Nevertheless, according to the back trajectories, the transport of mineral dust takes place directly

from the Sahara Desert situated in the south of the AMV site, without passing through cities with intense anthropogenic activities. The study of Decesari et al. 2010 reported similar concentrations of OC (0.8 ± 0.6 µg m$^{-3}$), and lower EC (0.1 ± 0.1 µg m$^{-3}$) concentrations in PM$_{10}$ at the Himalayan high-altitude station in Nepal. Furthermore, Sharma et al., 2020 reported higher OC (5.4 ± 2.0 µg m$^{-3}$) and EC (2.2 ± 2.0 µg m$^{-3}$) at the high-altitude site of Darjeeling, India most likely due to the higher influence of anthropogenic activities at the site.


The OC/EC ratio contributes to assessing the aging of aerosols during long-range transport as well as the impact of the combustion source producing EC. The average OC/EC ratio was computed as 4.8 ± 5.4 µg with a range of 0.2–56.1. Conversely, the OC/EC ratio tends to decrease from summer (11.2 ± 9.7) to winter (2.2 ±1.4). The decreasing trend of OC/EC ratio can be due to the formation of secondary organic aerosols in summer by

photochemical processes, as shown by the high secondary organic carbon (SOC) content observed during August (1.9 ± 0.8 µg m$^{-3}$) in contrast to December (0.1 ± 0.2 µg m$^{-3}$). At the same time, the increase of wood combustion from nearby urban regions during colder periods (autumn-winter) combined with the changes in the meteorological conditions, prevents the transport of pollutants (Chu et al., 2005). The highest OC/EC ratio (6.3 ± 7.5) was observed for MCE air masses, while the lowest ratio was recorded for BAM at about 2.2 ± 1.1, as shown

in Table 4. The OC/EC ratio observed at AMV for BAM was similar to those found in local samples in Northern Morocco with an average of 1.9 (Benchrif et al. 2018). Moreover, the OC/EC ratio shows a slight difference with those observed in Mt. Everest (Decesari et al., 2010) whose ratios varied from 5 to 9.  To conclude, carbonaceous aerosols show strong season and temporal variation with high OC values observed during summer due to biogenic emissions, mineral dust, and long-range transport, in contrast, to EC which shows low variability except during

few regional anthropogenic pollution events.

**3.3.3 Sea salt**

The Middle Atlas region is influenced by two maritime sources of sea salt, more often from the Atlantic Ocean, and sometimes from the Mediterranean Sea. During the study period, the average concentration of sea salt remains low (0.4 ± 0.5 µg m$^{-3}$) and contributed only 1.6% of the total PM$_{10}$ concentration. The highest concentrations were

recorded during August when sea salt concentrations reached a maximum of 3.4 µg m$^{-3}$. The sea salt then decreased gradually, reaching a minimum concentration of 0.06 µg m$^{-3}$ during December. The sea salt concentration was high when wind speed exceeded 6 m s$^{-1}$, indicating that sea salt is strongly dependent on meteorological conditions and air mass sources.

The higher concentrations of $Na^+$ (0.6 ± 0.3 µg m$^{-3}$) and $Cl^-$ (0.4 ± 0.3 µg m$^{-3}$) in the hot season (Aug-Sep) could be due to a larger contribution of marine aerosol. During this period, sea salt made up 11% of the total $PM_{10}$ mass, especially when the air mass came from the Atlantic Ocean (ACE) in comparison to < 1% of the $PM_{10}$ mass when the air masses were from the Sahara Desert (SD). However, no significant difference was noticed in sea salt concentrations found in the ACE (0.4 ± 0.7 µg m$^{-3}$) and MCE samples (0.4 ± 0.5 µg m$^{-3}$), as shown in Table 4. $Na^+$ concentration was high during a pollution episode on the 16$^{th}$ of August, which coincided with, high concentrations of EC, $SO_4^{2-}$, $NO_3^-$, and $NH_4^+$. This was due to the influence of ACE air masses with high EC content that made up about 2% of the $PM_{10}$ mass at the AMV site. The average ratio of $Cl^-/Na^+$ in $PM_{10}$ was found to be lower (0.4) than the value typically observed in seawater (1.8) (McInnes et al., 1994; Prodi et al., 2009). This points out to chlorine depletion due to chemical reactions that involve NaCl and $HNO_3$ or $H_2SO_4$ leading to the formation of $NaNO_3$ or $Na_2SO_4$ and gaseous HCl (McInnes et al., 1994). The estimated chlorine depletion was 42% for ACE and 49% for MCE air masses comparable with reported values from 28% to 63% (Avg. 48%) observed in $PM_{10}$ in the Atlantic Ocean and Mediterranean Sea (Contini et al., 2010, 2014). Consequently, sea salt was mainly present as aged sea salt at AMV during the sampling period. A detailed discussion on the correlation between $Na^+$ and $Cl^-$ will be elaborated on in section 3.4.

The comparison of sodium and chloride concentrations with other high-altitude studies is shown in table 5. The concentrations of $Na^+$ (1.8 ± 2.3 µg m$^{-3}$) and $Cl^-$ (0.8 ± 1.3 µg m$^{-3}$) are several times lower than those at Darjeeling in India which has a concentration of $Na^+$ and $Cl^-$, of 2.2 ± 2.0 µg m$^{-3}$ and 2.3 ± 1.5 µg m$^{-3}$, respectively. On the other hand, the concentration of $Na^+$ and $Cl^-$ were 4 to 8 times higher than the values reported at Mt. Everest station located at an altitude of 5079 m asl. In addition, the concentration of chloride was in good agreement with those observed in Mt. Cimone, Italy, 0.8 ± 0.9 µg m$^{-3}$. Sea salt concentration observed at the AMV (0.4 µg m$^{-3}$) was 5 times lower than at Tetouan (2.4 µg m$^{-3}$), a coastal Mediterranean city in northern Morocco, and approximately 20 times lower than Cap Verde Atmospheric Observatory (CVAO) located in the tropical Atlantic Ocean (Benchrif et al. 2018; Fomba et al. 2014). The contribution of marine aerosols originating from the Atlantic Ocean remains relatively low compared to other sites, but nevertheless existent.

### 3.3.4 Ammonium, nitrate, and sulfate

A significant part of PM composition was associated with the formation of secondary inorganic aerosols (SIA), which are mainly composed of sulfate, nitrate, and ammonium. They made up about 7.2% of the $PM_{10}$ mass. The temporal variation during the sampling period of $SO_4^{2-}$, $NO_3^-$, and $NH_4^+$ is presented in Fig. 6, with average concentrations of 0.9 ± 0.8 µg m$^{-3}$, 0.8 ± 0.6 µg m$^{-3}$, and 0.3 ± 0.2 µg m$^{-3}$, respectively. In summer, the concentrations were relatively high during few days in August, with the observation of the highest sulfate, nitrate, and ammonium concentrations of up to 6.1 µg m$^{-3}$, 4.4 µg m$^{-3}$, and 1.2 µg m$^{-3}$, respectively (Fig. 6). This was due to the transport of polluted MCE air masses through the Mediterranean Sea and across cities in the North of Morocco leading to high PM loaded aerosols. On average, the influence of long-range transport during the ACE and MCE air masses for sulfate (2.8 µg m$^{-3}$) and nitrate (2.3 µg m$^{-3}$) were similar. However, the contribution of ammonium (1.7 µg m$^{-3}$) to particulate matter was particularly higher for MCE air mass. Additionally, other peaks were also observed in aerosol concentrations both for $SO_4^{2-}$ and $NO_3^-$ during August. This could be attributed to the long-range transport of dust aerosol from the Saharan desert in Southern Morocco. The subsequent months

demonstrate a clear decreasing trend of SIA from high concentrations in summer (3.8 µg m$^{-3}$), to relatively low concentrations during winter (1.0 µg m$^{-3}$). In average, the sulfate concentration for MCE (1.2 ± 0.9 µg m$^{-3}$) was about 5 times higher than background sulfate concentrations (0.2 ± 0.2 µg m$^{-3}$). In addition, non-sea salt sulfate (nss-SO$_4^{2-}$) represented about 95% of the total SO$_4^{2-}$ at AMV and it had a strong correlation with NH$_4^+$, suggesting that secondary sulfate was mainly present as ammonium sulfate (detailed discussion later in section 3.4).

Secondary inorganic aerosol over the Atlas Mountains has been compared with the data reported in other high-altitude stations (Table 5). The average concentration of nitrate (0.8 ± 0.6 µg m$^{-3}$) at AMV was comparable with those reported in Mt. Himalaya and Mt. Cimone, 0.9 ± 0.2 µg m$^{-3}$ and 0.8 ± 0.7 µg m$^{-3}$, respectively (Chatterjee et al. 2010; (Marenco et al., 2006). However, the concentration of NO$_3^-$ were found to be approximately 2 times higher than the value reported in Puy de Dôme (Bourcier et al., 2012), as shown in Table 5. Two factors cloud be responsible for the high nitrate concentration recorded at AMV. Firstly, the mineral dust particles present in the Middle Atlas region contain calcium carbonates (calcite and dolomite) which can react with nitric acid gas in the atmosphere to form nitrate salts (Krueger et al., 2004; Khrissi et al., 2018). Second, the difference could be explained by long-range transport of polluted air mass from MCE and ACE which enhanced the nitrate concentration. Similar concentrations of ammonium at AMV (0.3 ± 0.2 µg m$^{-3}$) were found at Puy de Dôme, France (0.3 ± 0.2 µg m$^{-3}$), whereas the concentration was 5 times lower than those reported in other high-altitude sites such as the Mt. Himalaya (Chatterjee et al., 2010). This indicates that the influence of ammonium remains relatively low despite the proximity of the site to agricultural activities located in the surroundings of Meknes. The concentrations of sulfate (0.9 ± 0.8 µg m$^{-3}$) over AMV were comparable with those at Puy de Dôme (1.3 ± 1.1 µg m$^{-3}$), but were almost 4-5 times lower than all the other hilly stations except Mt. Everest (Decesari et al., 2010).

**3.3.5 Organic compounds**

The identification of the organic chemical compounds enables a better understanding of the organic fraction in the composition of aerosols and the quantification of the contribution of biogenic as well as anthropogenic emissions (Jaenicke, 2005). Therefore, a large number of individual organic chemical compounds were analyzed. Figure 6 shows the temporal variation of organic compounds, including n-alkanes, PAHs, n-alkan-2-ones and sugars.

*n-alkanes*

The distinguishing aspect of alkanes is their specific source and their ability to provide information about their origins (Pietrogrande et al., 2010). Individual n-alkanes with C-atom numbers in the range 19-34 were analyzed. Figure 6 shows the temporal variation of the n-alkanes revealing strong variations over the seasons with an average concentration of about 8.4 ± 7.1 ng m$^{-3}$. The average concentration decreases from summer (16.1 ± 8.9 ng m$^{-3}$) to winter (2.6 ± 2.0 ng m$^{-3}$). On average, the alkanes during BAM conditions was about 4.9 ± 3.2 ng m$^{-3}$ and was dominated by biogenic species such as heneicosane, hentriacontane, and nonacosane which made up about 60% of the total alkanes, as shown in Fig. S10. During SD air mass influence, such as in August, high concentrations of alkanes were observed (50.9 ng m$^{-3}$). The predominant compounds during dust events were pentacosane, hexacosane, heptacosane, and nonacosane suggesting that OC was loaded with biogenic matter (Pio et al., 2001).

MCE air mass influence revealed considerably high ($10.5 \pm 7.7$ ng m$^{-3}$) n-alkane concentration with elevated concentrations of typical anthropogenic tracers found on the samples such as nonadecane, and tricosane. This indicates that the long-range transport of MCE air masses was often loaded with anthropogenic material. While the contribution of MCE and SD air masses to alkanes is two times higher than during background conditions, the concentration recorded for ACE $5.6 \pm 3.7$ ng m$^{-3}$ remains relatively similar to BAM conditions. Additionally, organic compounds such as pristane and phytane considered as typical molecular markers of traffic emissions were rarely found in most of the samples. This indicates that the influence of traffic was considerably low.

To distinguish between natural biogenic emissions from plants and incomplete combustion, the carbon preference index (CPI) was also calculated and used as a marker (Alves et al., 2012; Pietrograndé et al., 2011). The part of n-alkanes with an even number of C-atoms exceeding the distribution of the average concentration of n-alkanes can be considered as coming from plant waxes. However, odd C-atom-numbers can originate from incomplete biomass combustion (Iinuma et al., 2007). Table 4 presents the CPI values calculated according to each air mass. The average CPI value was $3.8 \pm 2.4$ and ranged from 0.7 to 18.6. However, high CPI ($>>1$) was observed for all air masses, which indicates that the alkanes originated from plants waxes, as presented in Table 4 (Kavouras, 2002). In contrast, no values of CPI were recorded close to 1, which shows the minor influence of anthropogenic activities and traffic emission at AMV. The average concentration of alkanes was dominated by odd C-atoms with a concentration of 6.4 ng m$^{-3}$, compared to 1.9 ng m$^{-3}$ for even C-atoms. During summer, higher concentrations were observed, approximately 39.2 and 10.4 ng m$^{-3}$ for n-alkanes with odd and even C-atoms, respectively, Therefore, the average CPI increases during dust events during SD air mass influence during dust event ($4.0 \pm 3.1$) was due to the higher contribution of odd C-atoms alkanes. Similar to SD, the average CPI for MCE was about $3.9 \pm 1.9$ during MCE air mass influence was due to higher contribution of even C-atoms alkanes. However, a slight decrease of the mean CPI ($3.3 \pm 0.80$) was observed during BAM conditions occurring mainly during autumn and winter, as the biogenic activity was relatively low compared to the summer. Overall, the most dominant n-alkanes such as nonacosane and hentriacontane were observed during all the air masses indicating high local influence and they were typically associated with the biogenic activity.

*PAHs*

In the present study, polycyclic aromatic hydrocarbons (PAHs) with 3 to 7 rings were quantified. The temporal variation of the sum of the 20 identified PAH compounds is presented in Figure 6. The contribution of PAHs was much lower than alkanes with an average concentration of $0.6 \pm 0.8$ ng m$^{-3}$ over the whole study period. Contrary to what has been observed for alkanes, the PAH concentrations determined during the autumn months were higher than those during the summer and winter. The highest amount of PAH was detected during October, approximately 5.7 ng m$^{-3}$ due to long-range transport of MCE air masses, as shown in Figure 6. The minimum concentration was observed during winter, of about 0.05 ng m$^{-3}$. The average background concentration of PAHs was $0.4 \pm 0.5$ μg m$^{-3}$, which was low in comparison to other organic compounds likely because of high evaporation on warm days (Cincinelli et al., 2007). During MCE air mass, the PAH concentrations increased by 52% compared to the BAM concentration, as shown in Table 4. The most abundant PAHs found in the BAM samples were fluorene and retene, which represent 75% of total background PAH concentrations. The abundance of fluorene (Fa) and retene (Rete) were found in samples from different air masses suggesting that they potentially originate from similar local or regional

emission sources. Moreover, fluorene and retene are marker compounds for wood combustion or combustion of organic substances, but it is also found in trace concentrations in the combustion of gasoline or diesel (Spindler et al., 2012). Other compounds such as coronene, dibenzo(ah)anthracene, or phenanthrene were observed during long-range transport of polluted air masses during MCE air mass influence. The contribution of PAHs was lower for ACE (0.4 ± 0.4 µg m$^{-3}$) air masses than for MCE air mass influence, which indicates that not all long-range transport was loaded with combustion tracers. The average PAH concentration was higher during SD than ACE air masses of about 0.7 ± 0.7 µg m$^{-3}$. The most abundant compounds were fluorene and 9H-fluorenone, which were found within a similar concentration range. Therefore, mineral dust transport was not affected by the combustion processes. As a result, the observation of PAH concentration shows a strong variation with high biogenic activities in the surroundings during summer and high anthropogenic PAHs during pollution episodes from combustion processes in autumn.

*n-alkan-2-ones*

In total, 5 n-alkan-2-ones were detected in this study, as shown in Fig. S8. The n-alkan-2-one concentrations increased significantly from summer (1.8 ng m$^{-3}$) to autumn (9.7 ng m$^{-3}$), then decreased continuously to winter (6.3 ng m$^{-3}$), with an average of 6.6 ng m$^{-3}$ for the whole sampling period. The minimum concertation was recorded during the summer of about 0.6 ng m$^{-3}$. In contrast, the maximum concentration was reached during autumn of about 52 ng m$^{-3}$ due to ACE air mass influence, as shown in Fig. 6. The sum of n-alkane-2-one was between 0.67 to 13.2 ng m$^{-3}$. The same relative composition of n-alkan-2-one concentrations was observed in both seasons, suggesting that they came from similar sources. However, the levels of n-alkan-2-one were much lower in concentration than those of n-alkanes. The average background concentration of the total n-alkan-2-one was 5.9 ± 5.5 ng m$^{-3}$. During this period, major n-alkan-2-one constituents recorded at AMV were 2-Nonadecanone, 2-Heptadecanone, 2-Octadecanone, which represented 29%, 25%, and 18%, of the total, detected n-alkan-2-one, respectively. These organic compounds that appear in fossil fuel burning events were found in low concentrations at the AMV. The remaining part was made up of 2-hexadecanone and 2-octadecanone, which made up for 15.6%, and 6.2%, of the total n-alkan-2-ones, respectively. The discrepancy of n-alkan-2-ones concentrations in comparison to BAM chemical composition was low from August to October. The trend is completely reversed, as the average concentration was significantly elevated during November and December. During these two months, the concentration increased by 35%. Some spikes detected in November and December are characterized by a change in wind direction, and high wind speeds. These samples, with higher concentrations for 2-heptadecanone and 2-nonadecanone show a strong correlation with temperature. Indeed, this period marks the beginning of winter, with average temperatures dropping as low as 4,5 °C, as shown in Table 1. No correlation between alkane-2-ones was found with elemental carbon, but the analysis of the data shows a strong correlation only for the night samples with As and K$^+$. This indicates that a possible source of the n-alkan-2-ones is combustion due to residential heating. These data present the first measurement made that shows the influence of combustion in North Africa on n-alkan-2-ones concentrations. Similar conclusions were reported by Müller, (1997) who highlighted the anthropogenic source of alkane-2-ones during winter.

Two main sources are known to be responsible for the presence of n-alkane-2-one in the air: Incomplete combustion and in-situ microbial α-oxidation of the carbon chain (Khedidji et al., 2020). The first source often creates n-alkane-2-one with a predominance of odd-numbered carbon atoms; on the other hand, the last source gives rise to the opposite. In the present study, a strong predominance of odd n- alkane-2-one rather than pairs was observed, suggesting that the incomplete combustion of organic material was the principal ambient source in

this region. In particular, heptadecane-2-one (K17) was the most abundant in all the samples and reached a maximum 5.2 ng m$^{-3}$.

 *Sugar alcohols*

Three main sugar alcohols that are, levoglucosan, arabitol and glucose were identified between August 2011 and December 2017 in the AMV samples. The sugar concentration levels in the aerosol samples ranged from 0.02 to 39.6 ng m$^{-3}$. The average concentrations of sugar compounds were higher during summer (7.9 ng m$^{-3}$) and decreased continually until November (1.2 ng m$^{-3}$). During December, the concentrations were relatively higher than November (2.1 ng m$^{-3}$) where some high peaks were observed. Glucose was about five times higher in summer than in winter. Notably, there were three extreme days where high sugar concentrations were observed, as shown in Fig. 6; 13$^{th}$ August 2017 (39.6 ng m$^{-3}$), 18$^{th}$ September 2017 (ng m$^{-3}$), and 20$^{th}$ October (28.6 ng m$^{-3}$). These peaks are due to the long-range transport from the coast of Europe (ACE) and the coast of the Mediterranean Sea (MCE). The average sugar concentrations during these air mass influences were ACE (3.7 $\pm$ 6.0 ng m$^{-3}$) and MCE (5.2 $\pm$ 7.7 ng m$^{-3}$), as listed in Table 4. In contrast, sugar compounds were relatively low in SD (1.8 $\pm$ 2.9 ng m$^{-3}$) air masses and were not found in the background PM$_{10}$ conditions. Levoglucosan which is considered as a good tracer of biomass burning emissions in aerosol particulate matter, was particularly higher for ACE (2.0 ng m$^{-3}$) and MCE (1.6 ng m$^{-3}$) air mass influence, as displayed in Fig. S10. (Bauer et al., 2008). Arabitol shows a similar concentration for MCE and ACE with a mean of 1.0 ng m$^{-3}$ suggesting that particles were loaded with primary biological aerosols such as pollen, fungal spores, vegetative debris, viruses, and bacteria from the marine coast (Fu et al., 2012). Glucose remained relatively high during MCE air mass influence in comparison to other air masses influence. During SD air mass influence, the concentration of arabitol was extremely low with a concentration less than 0.08 ng m$^{-3}$. However, glucose showed a higher concentration of about 0.7 ng m$^{-3}$ but remains 3 times lower than MCE concentrations. This indicates that the sugars were most likely originated from marine air masses.

To conclude, sugars have potentially two major sources at AMV: a natural biological source from marine air masses including MCE and ACE and an anthropogenic source from biomass burning potentially from urban cities close to the site. The contribution of arabitol and glucose was significantly higher during the summer, linked to more developed vegetation and higher biogenic activity, and in contrast to winter, levoglucosan was higher.

### 3.3.6 Crustal enrichment factor

Analysis of the crustal enrichment factor (EF) has been used to estimate the contributions of crustal matter to the ambient PM$_{10}$ particles at AMV. For this study, titanium (Ti) was used as a reference element due to the low recovery of Al and high recovery of Ti, and as it is also considered a suitable tracer for mineral dust (Fomba et al., 2013). Furthermore, Al or Fe has more anthropogenic sources than Ti. However, the comparison of EFs for Ti and Al as reference elements shows a similar trend, with only slight differences observed in the absolute values. The average upper continental crust composition, according to Wedepohl (1995) was used for the calculation of the enrichment factors. The EF relative to Ti was calculated using equation 2 as follow:

$$EF = \frac{\left(\frac{Z}{Ti}\right)sample}{\left(\frac{Z}{Ti}\right)Crust}$$

(2)

The enrichment factor provides the ability to classify metals based on their enrichment to the soil. Elements with an EF under 2 are considered to have a similar composition to the reference soil values. An enrichment factor above 2 but below 10 is assumed to have low enrichment with a possible mixture of both crustal and non-crustal sources. Elements with an EF above 10 are considered enriched, while enrichment factors above 100 are considered highly enriched, suggesting that the elements are from non-crustal and more likely anthropogenic sources. The enrichment factor does not take into account each pollution episode but is a general approach to the classification of metals according to their crustal origin. Within the present study, the elemental enrichment factors showed similar trends for the different air mass inflow to the station. Three groups of elements could be identified from the elemental enrichment factors. Figure 7 shows the average PM10 elemental crustal enrichment factors (at AMV according to the respective air mass origins.

Group I includes elements such as Al, Ba, Rb, K, and Fe with enrichment factors between 0.8 and 2. Their enrichment factors suggest that these elements are associated with particulate matter from the resuspension of soil or other crustal sources. Elements such as Al, Fe, and Mn show little dispersion, and their variation seems to be constant across different air masses, clearly indicating that the source was soil. As suggested by other studies, these metals could also have an anthropogenic source, but in this study, they clearly showed crustal matter origin (Viana et al., 2008; Birmili et al., 2006; Contini et al., 2012). On one hand, no correlation was found between Al, Fe, and anthropogenic tracers such as EC or other heavy metals, which indicates their natural origin. On the other hand, K showed slightly higher EFs for air masses from the Atlantic Ocean, suggesting that sea salt and sources other than mineral dust, such as biomass combustion, might have contributed to its presence.

Group II elements include heavy metals such as Sr, Ca, Cu, Mn, Ce, V, La, Co, and As. These elements had enrichment factors ranging from 2 to 10, indicating the possibility of having mixed origin from both crustal and anthropogenic sources. The lowest enrichment factors were observed during BAM suggesting that the elements may have been of crustal origin. In contrast, the highest enrichment factors were mainly observed in air masses that originated from the coast of Europe (ACE) and crossed major urban cities such as Rabat/Salé/Kenitra and Casablanca before arriving at Atlas station M5. In this case, it is assumed that these elements were probably influenced during transport by anthropogenic emissions. In contrast, the Mediterranean Sea air mass appears to remain relatively unaffected by anthropogenic emissions. In addition to its atmospheric crustal origin, V had a high enrichment factor mainly due to residual oil combustion, especially at night. Particles from oil combustion processes were often observed in high concentrations during winter due to their size and long lifetime in the atmosphere and the combustion activities in the nearby urban cities.

Group III contains the elements with EF from 10 to 1000, including heavy metals such as Cr, Zn, Ni, Pb, as well as Br, Se, and Sb. These elements showed high enrichment factors in all air mass directions. They are mainly present in the marine air masses of the Atlantic, but also the Mediterranean air masses. An increase in heavy metal

concentration has been observed during winter, and at night when the temperature drops and the air mass inflow

from the cities towards the mountain prevails. Atmospheric Ni and Cr are released during combustion processes, while Pb is mainly released from smelters or the combustion of unleaded petrol, waste, and coal (Pacyna et al., 2007). Combustion processes are generally the main contributors to these anthropogenic metals. Zn had a weak correlation with Pb and Ni, suggesting that its origin is also anthropogenic. The nearest urban cities are Meknes

and Fes, where anthropogenic activities such as waste incineration, and road traffic pollution are common. Furthermore, the V/Ni ratio was observed higher for MCE air mass, about 2.8 which is considered typical for heavy fuel oil combustion (Mazzei et al., 2008; Pandolfi et al., 2009; Bove et al., 2014).

### 3.4     Inter-relationship between aerosol components

The inter-relationship between the different species and the scatter plots are presented in Fig. 8. The analysis of

the single correlation coefficients allows obtaining information about the possible common sources of aerosol.

### 3.4.1 Nitrate and nss-sulfate

The correlation between $NO_3^-$ and nss-$SO_4^{2-}$ ($r^2$=0.76) indicates their possible common origin. The correlation was more pronounced for MCE air masses ($r^2$=0.80) in contrast to ACE ($r^2$=0.43) air masses, suggesting an enhanced transport of secondary anthropogenic aerosol from the Mediterranean coast (Liu et al., 2017) to the

AMV site. The nss-sulfate concentrations were slightly correlating with vanadium which is associated with the emissions of oil combustion, ship emissions as well as iron and steel industrial emissions (Pandolfi et al., 2011). A strong correlation of $NO_3^-$ and nss-$SO_4^{2-}$ was observed with oxalate ($C_2H_4^{2-}$), which could indicate that they have a common source and that they can originate from biomass burning and secondary transformations. Nss-$SO_4^2$ also originated from crustal sources especially as elevated concentrations were observed during dust events.

This assertion was supported by a good correlation of nss-$SO_4^{2-}$ with nss-$Ca^{2+}$ (Fig. S7), indicating the likely presence of calcite particles of crustal origin. A similar observation was reported by Okada and Kai, (2004), who observed that Desert dust was associated with sulfur compounds and organic matter from surrounding agricultural areas. Indeed, the particles with high sulfate content were accompanied by Ca and were assigned as gypsum particles, also suggesting that the sulfur in these particles originated from a sedimentary source (Falkovich et al.,

2001).

### 3.4.2 Ammonium nitrate and ammonium sulfate

The analysis of the correlation matrices between nss-$SO_4^{2-}$ and $NO_3^-$ with ammonium ($NH_4^+$) was applied to better understand the inter-relationship between the secondary inorganic species. A correlation between nss-$SO_4^{2-}$ and $NH_4^+$ ($r^2$= 0.90) supported the hypothesis of dominant ammonium sulfate particles $(NH_4)_2SO_4$ in the summer

especially when air masses were coming from ACE, as shown in Fig. 8. During this period, a strong correlation was found between sulfate and solar radiation which suggests that nss-$SO_4^{2-}$ was produced via photochemical reaction (Baker and Scheff, 2007). Nevertheless, the transport of nss-$SO_4^2$ from the Atlantic coast also contributes to the formation of ammonium sulfate. However, the trend is more towards ammonium nitrate ($NH_4NO_3$) in winter, given that the main correlation of $NH_4^+$ with $NO_3^-$ ($r^2$= 0.95) mainly present in MCE air masses. Nitrate

shows a strong dependency on the temperature at AMV, most likely due to the stability of ammonium nitrate in the atmosphere at low temperatures (Squizzato et al., 2013). The predominance of nitrates over sulfates during winter, where nitrates and ammonium remain high, is probably due to the influence of temperature that prevents

the dissociation of ammonium nitrate particles (Ricciardelli et al., 2017). Moreover, a similar pattern of $NO_3^-$ and $NH_4^+$ as observed by Querol et al., 2004 in the Mediterranean coast with a summer minimum and suggested that
it could be due to the low thermal stability of the nitrate in the hot season.

### 3.4.4 Sodium and chlorine

The evolution of the sea salt constituents and their relationship with the most important aerosol acidic species such as $NO_3^-$ and $SO_4^{2-}$ was investigated according to their air mass origins (Fig. 8). A correlation between sodium and chlorine was observed ($r^2$=0.76), as shown in Fig. 8. The scatter plot of molar equivalent concentrations of
$Na^+$ and $Cl^-$ shows a strong correlation specifically for ACE and SD air masses. However, the data points are below the seawater reference line and only approach this line when the $Cl^-$ concentration is combined with $NO_3^-$ and nss-$SO_4^{2-}$. This indicates that chloride was depleted in the sea salt particles due to the displacement of chloride by sulfate from sulfuric acid when air masses were coming from MCE and ACE, especially as photochemical processes favor sulfate formation during summer. The same scenario has been observed for $NO_3^-$ with a
considerable difference during the winter. Indeed, the correlation between sodium and the sum of chloride and nitrate shows the chloride depletion and indicates that the Mediterranean Sea air mass was loaded with aged sea salt. Similar results were observed in the North of Morocco where the mass fraction of nitrate was higher in the coarse fraction which indeed corresponds to aged sea salt (Benchrif et al. 2018). No correlation between $Na^+$ and $Cl^-$ was observed in the BAM conditions.

### 3.5    Day and night-time variation

*Inorganic ions, mineral dust, and organic carbon (OC)*
Diurnal variations of various $PM_{10}$ chemical species were analyzed to understand the influence of day and night variations on their concentrations. Figure 9 shows the variation for given chemical species. The OC increase from night-time (0.9 μg m$^{-3}$) to the day (1.2 μg m$^{-3}$) and was accompanied by a slight increase in alkanes such as
pristane, docosane, and nonacosane, as observed in Fig. 9a. These alkanes indicate that the organic fraction was dominated by biogenic sources during the day. The highest concentrations of biogenic compounds were reached during the summer. During summer, higher concentrations of Al (0.6 μg m$^{-3}$) and Fe (0.7 μg m$^{-3}$) were observed during the day, compared to the night. This increase seems to be related to an additional source from the resuspension of road dust, due to car traffic during the day. Nevertheless, the transport of mineral dust tracers
from the Saharan dust could be controlled by other factors. For example, the study done by Khan et al., (2015) indicates that the penetration of dust into the free troposphere in the Atlas Mountains can also be due to orographic lifting, convection on the mountain slopes, and updrafts in the breeze front. While the $Ca^{2+}$ concentration which is approximately 0.6 μg m$^{-3}$ seems stable between day and night. The continuous presence of calcium indicates that it comes from a different source most likely local or regional. This difference in trace metal concentrations
shows the important role that calcium plays as a local source.

*Elemental carbon (EC), anthropogenic metals, and PAHs*
The composition of $PM_{10}$ in the evening is characterized by a high concentration of anthropogenic trace elements. The EC concentrations slightly increased in the evening from 0.2 to 0.3 μg m$^{-3}$. Besides, an increase of PAHs and
alkane2-one concentrations such as fluorene, retene, and 2-nonadecanone was observed during the night-time. The PAH concentrations during the day and night-time were 0.6 and 0.8 μg m$^{-3}$, respectively. In particular, the

Fluorene was the most abundant PAH in all the night samples and reached a maximum of 2.6 ng m$^{-3}$. A correlation ($r^2$=0.67) was found between PAHs and EC during night-time indicating their anthropogenic origins. In addition, anthropogenic metals such as Pb, Cr, V, Cr, Ni, and Cu associated with combustion and traffic emissions, increase by a factor of 1.8 during the evening (Fig 9b). Indeed, the anthropogenic influence at the AMV site occurs during the evening due to two important factors. First, the site is in a mountainous region influenced by the temperature fluctuation between day and night. The rapid cooling between day and night was accompanied by a change of aerosol sources. This phenomenon is widespread, especially in summer. Second, the variation of the air mass, combined with a change in the height of the boundary layer, contributes to the transport of pollutants from urban sites Fes and Meknes to the AMV site.

*Influence of meteorology*

The variation of the meteorological parameters between day and night is a critical factor that can indeed influence the chemical composition of the particles. First, a significant difference in PM$_{10}$ was observed on days when the day and night temperature difference was substantial, for example, on the 12th of August. The concentration decreased during the day (113 µg m$^{-3}$) to 80.4 µg m$^{-3}$ at night. Second, the influence of meteorology on secondary inorganic aerosols, such as sulfate and nitrate, were characterized by different variations between day and night. On the one hand, the sulfate is slightly enhanced from night (0.9 µg m$^{-3}$) to the day (1.2 µg m$^{-3}$). This increase during the day could be explained partly by sulfate originating from dust resuspension. However, a correlation between solar radiation and sulfate suggests that photo-oxidation during the day could also be a source of the sulfate increase. Whereas, nitrate recorded during the day (0.8 µg m$^{-3}$), shows higher concentrations during the night especially (1.0 µg m$^{-3}$). The drop in temperature between the day and the night, especially during the winter, allows for a rapid formation of ammonium nitrate.

*Mechanism of day-night variation*

In principle, two mechanisms control the variation between day and night: The wind direction and the boundary layer height. The wind direction plays an important role because it introduces air mass transported from different sources. Indeed, the winds blows from all directions, but it is dominated from the east section during the day, and by the west during the night, as shown in Fig. S10. High speeds were recorded during the night, up to 17.5 m s$^{-1}$, mostly associated with marine air masses. This suggests that long-distance transport often occurred during the night while the wind speed during the day was relatively lower. The lower wind speed during the day indicates that the influence of local sources is important. Furthermore, the topography, as well as the embedded valleys, also play a role in the pollution transport during the daytime as shown by Lang et al., (2015). Mountains can give rise to daytime upslope winds and night-time downslope winds. The valley bottom warms during the day, warm air rises the slopes of the surrounding mountains and hills to create a valley breeze. During night-time, radiation from the earth's surface cools the slopes, causing cooler, denser air to drain into the valley. In addition, local boundary layer processes and long-range transport contribute to chemical composition changes (Nair et al., 2007). A similar impact of mountain-valley circulations on air pollution was observed by Bei et al. (2018). Studies at high-altitude sites in southwest India, also found that diurnal variations of aerosol particle concentrations were related to mountain valley winds and the variation in a planetary boundary layer height (Buchunde et al., 2019).

### 3.6    Differences in chemical composition between dust and non-dust events

To investigate the impact of the dust event on the $PM_{10}$ chemical composition, the data has been segregated into two categories dust and non-dust episode. Only selected days with high influence of Saharan air mass with Al > 1 µg m$^{-3}$, were representative for dust event days.  This approach was previously used by Koçak et al., 2012 to investigate the influence of mineral dust in the eastern Mediterranean. Whereas the non-dust samples were categorized based on aerosol Al concentrations (Al < 1 µg m$^{-3}$). All the samples that constitute a mixing scenario were excluded. Figure 10 shows the average concentrations of (A) $PM_{10}$ mass, (B) Fe/Al and Fe/Ca ratios, (C) OC and major ionic species, (D) EC and minor ionic species, (E) organic compounds during non-dust and dust events.

*Mass*

The lang-range transport of mineral dust showed a significant impact on $PM_{10}$ composition. During dust events, $PM_{10}$ concentrations were on average 3 times higher, and up to a maximum of 10 times higher, in comparison to non-dust days. Mineral dust was about 5 times higher in comparison to non-dust samples (Fig. 10a). In addition, RH was lower during dust event days ranging from 20% to 45%, whereas it was 50% to 70% on the non-dust days. Similar results were observed by (Mukherjee et al., 2020) showing the impact of dust on the local meteorological conditions.

*Minerals, metals, and ions*

Aerosol inorganic species demonstrate distinct differences in chemical composition between dust and non-dust events. Although the North African mineral dust is mainly made up of clay minerals and quartz, the content of calcium carbonates varies depending on the North Africa source (Chiapello et al., 1997; Glaccum and Prospero, 1980). A comparison between Fe/Al and Fe/Ca ratios were used to provide the potential geographical origin of mineral dust according to their chemical composition (Formenti et al., 2014). The variability of the Fe/Al ratio was relatively low (0.9) during non-dust events and decreased to about 0.6 during the dust events, as shown in Fig 10b. The Fe/Ca ratio was also used to make distinctions amongst sources. It was found that on average, the Fe/Ca ratio was 1.4 during dust events, and 0.4 during non-dust events. These ratios, which are robust indicators of large-scale mineral dust source variation, were supported by air mass backward trajectories.

The back trajectory analysis indicates that particles during dust events came from the Saharan region of Mauritania and southern Morocco as also highlighted by their high Fe/Ca ratio while mineral dust during BAM as described above was of local sources emitted from road-dust or resuspension from agricultural activities. Fe/Ca ratios close to 0.4 indicate that the dust comes from the south, while Fe/Ca ratios higher than 1 indicate that the dust comes from the East. The average Fe/Ca ratio obtained at the station was 0.54, which suggests that the dust often originated from south Morocco. During the dust event, the Fe/Ca ratio reached 1.9, which suggests that long-distance transport of the dust from the eastern Saharan regions was observed, which agrees with the pollution rose, which indicated high concentrations from southeast winds.

An increase in the concentration of many ions was also observed during dust events. Sulfate, nitrate, calcium, ammonium, showed an increase in the average concentration of about a factor of 4.5 while chloride, magnesium,

and potassium experienced an increase in their concentrations by a factor of 5 (Fig 10c, 10d). During dust events, sodium concentration also experienced an increase by a factor of 3. Its correlation with iron and aluminum suggested its possible soil origin. Other metals of anthropogenic origin (not plotted), such as Cu, Ni, and Pb, showed no significant difference between dust and non-dust events.


*Organics*

Figure 10c reveals that organic carbon increased averagely during the dust event from 0.5 to 3.5 µg m$^{-3}$. Samples collected during the dust period showed a strong correlation between organic matter and other elements of crustal origin such as nss-Ca$^{2+}$ and nss-Mg$^{2+}$. The OC/EC ratio was in the range of 4-6 during the dust period, while the

ratio was lower than 3 during BAM which suggests that the organic fraction was affected by desert dust particles. Specifically, there is a clear correlation between Fe, Mn, Al, K$^+$, Ca$^{2+}$ and OC to a less extent, Mg$^{2+}$. During this period, a correlation was observed between OC as well as some organic species nonacosane and heptacosane with mineral elements such as calcium (r$^2$=0.81) and magnesium (r$^2$=0.73). During the winter, this relationship became practically insignificant (r$^2$=0.15). This suggests a possible common origin of this species that is the crustal

mineral aerosol as it is also confirmed by the low enrichment factor of all the species mentioned. Some organic compounds increase during dust events, especially odd alkanes such as nonaconsane, hentriacontane, heptacosane, and tricosane by a factor of 4. In contrast, elemental carbon remains globally constant with marginal changes (Fig 10e). The PAH fluorene, showed similar concentrations during dust events and background conditions, indicating that Saharan dust was not a significant source. An increase of about 65% was observed for pentacosane,

octacosane, hexacosane and decosane. Due to the biodiversity of several plant species and remarkable microbiological activity surrounding the Atlas regions, an increase of some organic compounds and organic matter fraction suggests that mineral dust was loaded with biogenic compounds during dust transport.

**4    Conclusion**

In the present study, PM$_{10}$ particulate matter was chemically characterized at the newly established AMV research

station located in the Middle Atlas region (Morocco) at an altitude of 2100 m from August to December 2017. The aerosol chemical composition was evaluated during remote background conditions and the main air mass origins were identified. The data shows an overview of the background chemical composition and the different sources affecting aerosol composition at such a remote high-altitude site. The influence of desert dust was investigated as the site location is close to the Sahara Desert.

Despite the proximity of the site to the Saharan Desert, the influence of the desert on the atmospheric composition at this altitude was only seasonal. PM$_{10}$ mass concentration showed a decreasing trend with high concentrations during summer due to dust events and significantly reduced during autumn due to the washout effect from enhanced rainfall. Four main air mass inflows at the site were identified using back trajectory analysis, with each air mass distinguished by different chemical compositions. The influence of marine air mass from the

Mediterranean Sea is prevalent at AMV and made-up 37% of all air masses. The chemical composition in the Middle-Atlas during sampling period is mainly dominated by locally emitted dust (61%) with high contribution from road dust, ionic species (7%), organic matter (7%), water content (12%), and indeterminate mass (11%). Biogenic organics contributed up to 7% of the organic matter. Organic matter increased during dust events due to biogenic crustal material emissions. Diurnal variation of PM was related to the variation in a planetary boundary

layer, mountain-valley winds as well as changes in different local sources. Mineral dust influenced was seasonal and ranged between 20 and 74% of the mass concentration on $PM_{10}$ with peaks observed during the summer, accompanied by high concentrations of $SO_4^{2-}$ of up to 3.0 µg m$^{-3}$.

During winter, $PM_{10}$ concentrations are low, the influence of the desert is weaker, and the marine air masses are more dominant with a mixture of polluted aerosol from the coastal regions of Rabat and Casablanca and sea salt's observed. High concentrations of mineral dust were observed during the daytime due to the resuspension of road dust, while an increase of PAHs and anthropogenic metals such as Pb, Ni, and Cu were found during night-time because of the boundary layer variation. Data show that proximity to the desert does not necessarily imply constant exposure to mineral dust. Furthermore, topography and temperature variation at mountain sites control PM
concentrations.

This is the first high altitude aerosol characterization study in North Africa which fills an important gap in the African region presently not available. The data from the AMV sites thus present a reference for aerosol particle composition under regional background conditions as well as during the influence of continental air masses.
Several other studies are needed to better understand the influence of the desert on the chemical composition but also the microphysical properties in the Middle Atlas region. In this study, only the chemical composition of bulk $PM_{10}$ particles was investigated, however, the size distribution remains an important factor and should be considered in further studies. An additional study on the chemical composition of the urban cities near the AMV, especially in Fes, would allow a better understanding of the anthropogenic influence in the North of Morocco.


*Data availability.* All data will be made available upon request by the authors.

*Author contributions.* WM, HH, and SE designed the experiment at the AMV station, KWF and ND performed the fieldwork, collected the samples, and performed the data analysis. ND performed the laboratory investigations, compiled the final Figures, and wrote the article. LP has contributed to the back trajectories calculation and the classification of air masses. ND and KWF undertook the results interpretation to which HH contributed. SZ contributed to text improvement for certain sections of the manuscript. All authors reviewed, edited, and contributed to the article.

*Competing interests.* The authors declare that they have no conflict of interest.

*Acknowledgements.* The authors would like to thank Ibrahim Ouchen, Sayf El Islam Barcha, and Mehdi El Baramoussi for assistance in sample collection at the AMV. The authors would also acknowledge the support of Kangwei Li, Julia Wilk, Susanne Fuchs, Sylvia Haferkorn, and Cornelia Pielok for their support with the trace metal analysis.

*Financial support.* This research has been supported by the European Union's Horizon 2020 research and innovation programme (MARSU, grant no. 690958)

The publication of this article was funded by the Open Access Fund of the Leibniz Association.

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

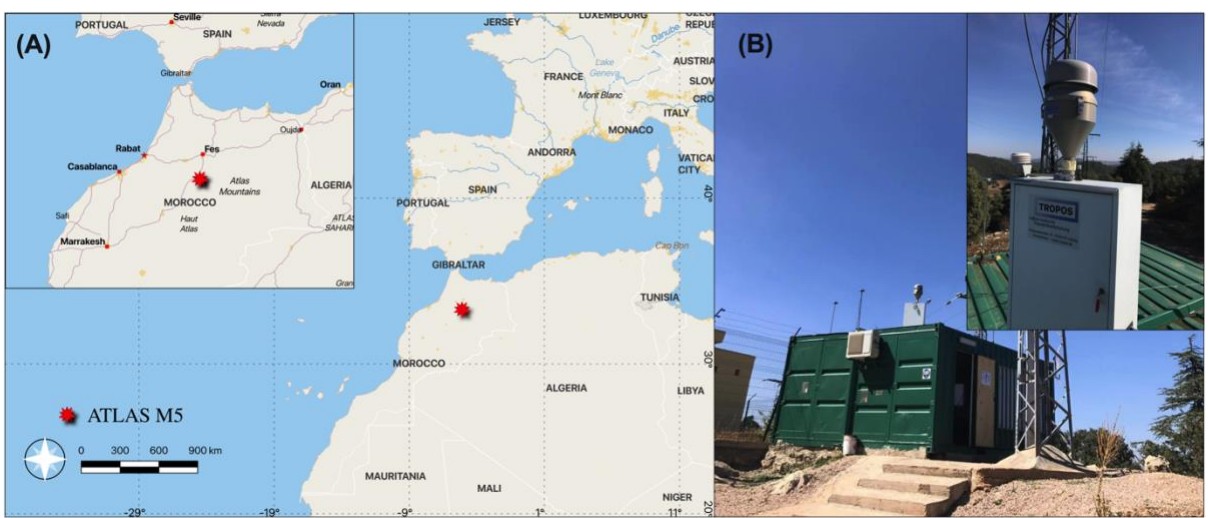

**Figure 1.** (A) Location map of AMV site in the Middle-Atlas; (B) Photo of the AMV site.

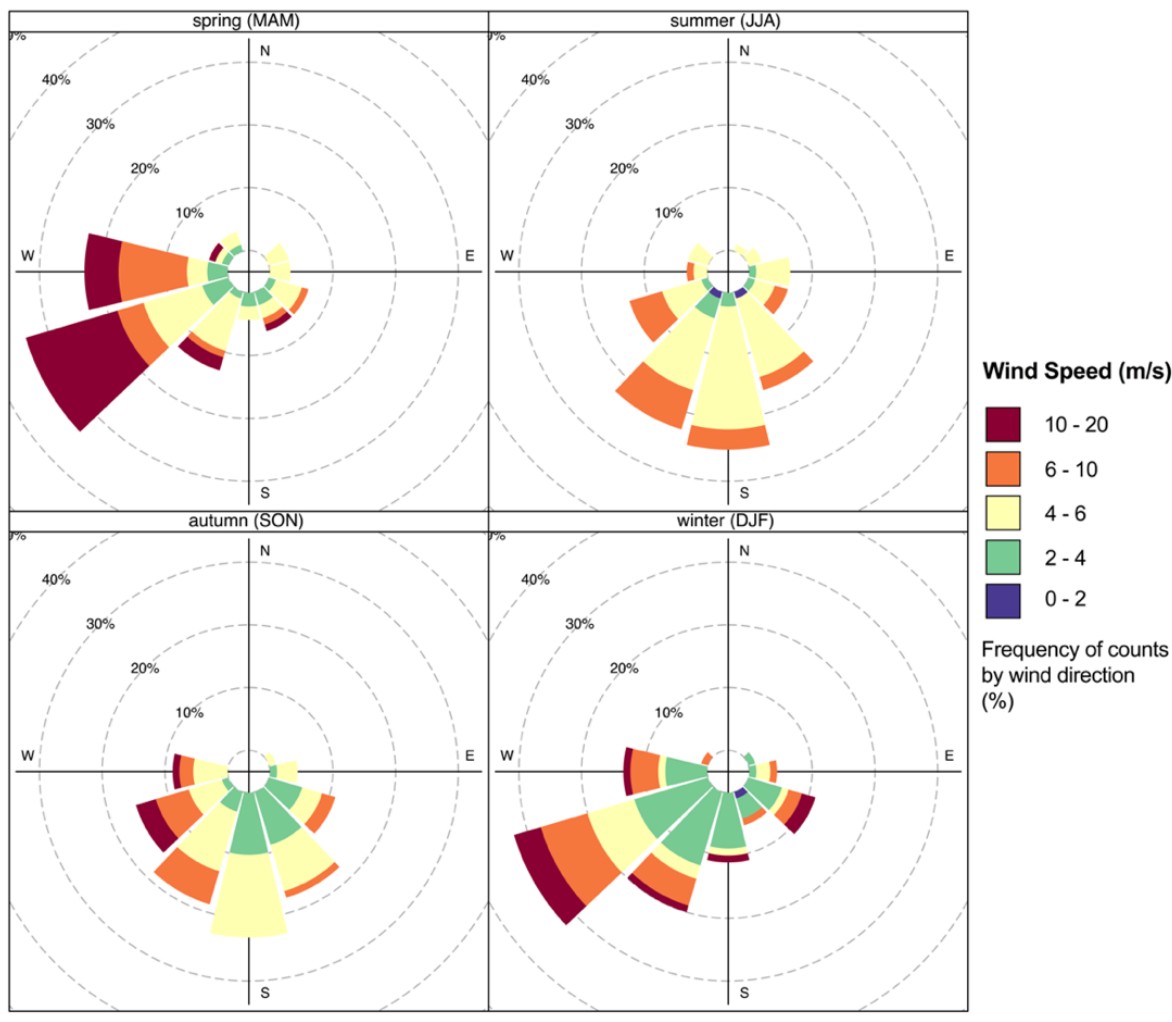


**Figure 2.** Seasonal wind rose plots at AMV.

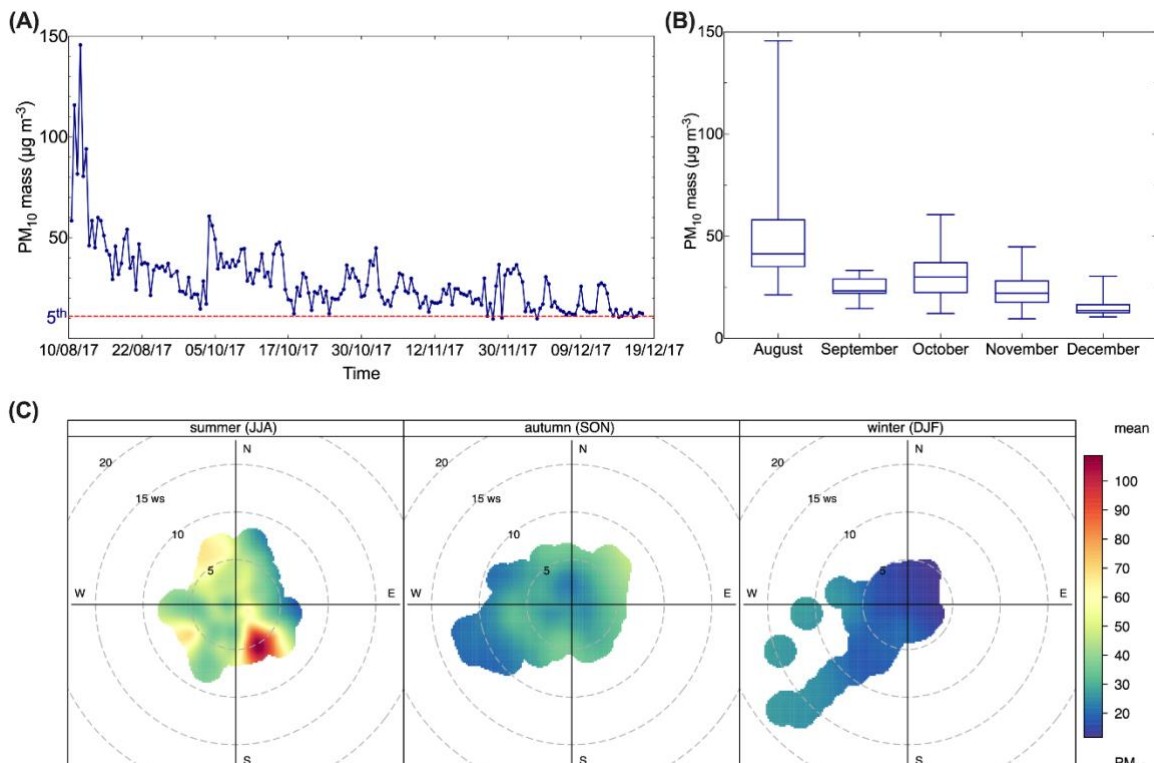

**Figure 3.** (A, top right) Time series of daily $PM_{10}$ mass; (B, top left) Box plot of monthly averages of $PM_{10}$ mass; (C, bottom) Pollution rose of $PM_{10}$ mass; The presented data were separated according to each season into summer (Aug), Fall (Sep-Nov), and winter (Dec).

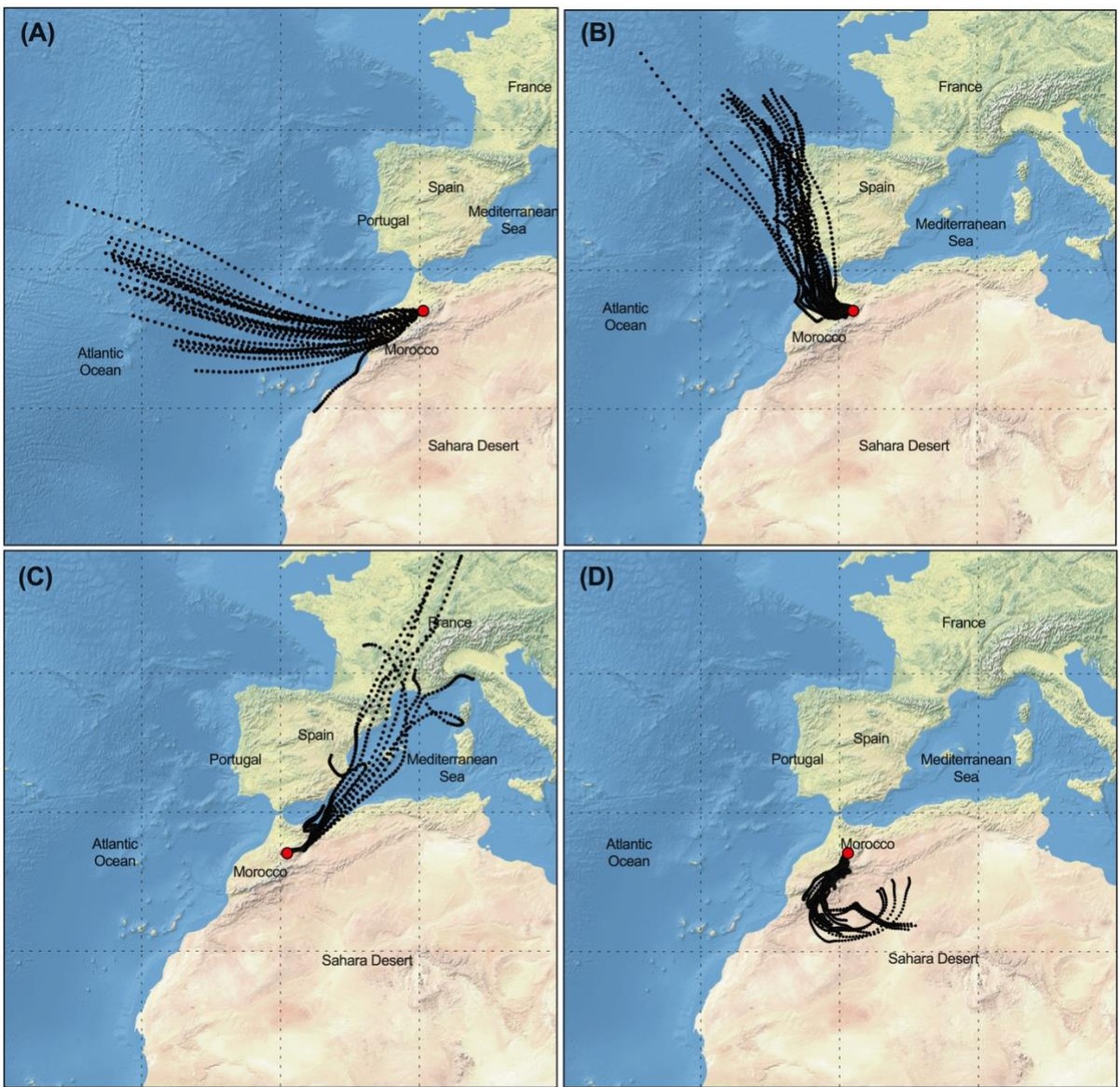


**Figure 4.** Typical 96h air mass back trajectory performed for AMV during routine samples periods; aerosol type and $PM_{10}$ mass concentration are given in parentheses: (a) 18 December 2017: air mass from the North Atlantic Ocean considered to be representative of background conditions (BAM, $m$=10.9 µg m$^{-3}$); (b) 10 October 2017: air mass from Europe crossing coastline of North Morocco (Atlantic Coast Europe, $m$=44.1 µg m$^{-3}$); (c) 2 November 2017: slightly polluted air mass from North East crossing Mediterranean Sea Morocco (Mediterranean Coast Europe, $m$=26.4 µg m$^{-3}$) ; (d) 13 August 2017: dust loaded air mass coming from Sahara desert (Saharan Dust, $m$=94.1 µg m$^{-3}$).


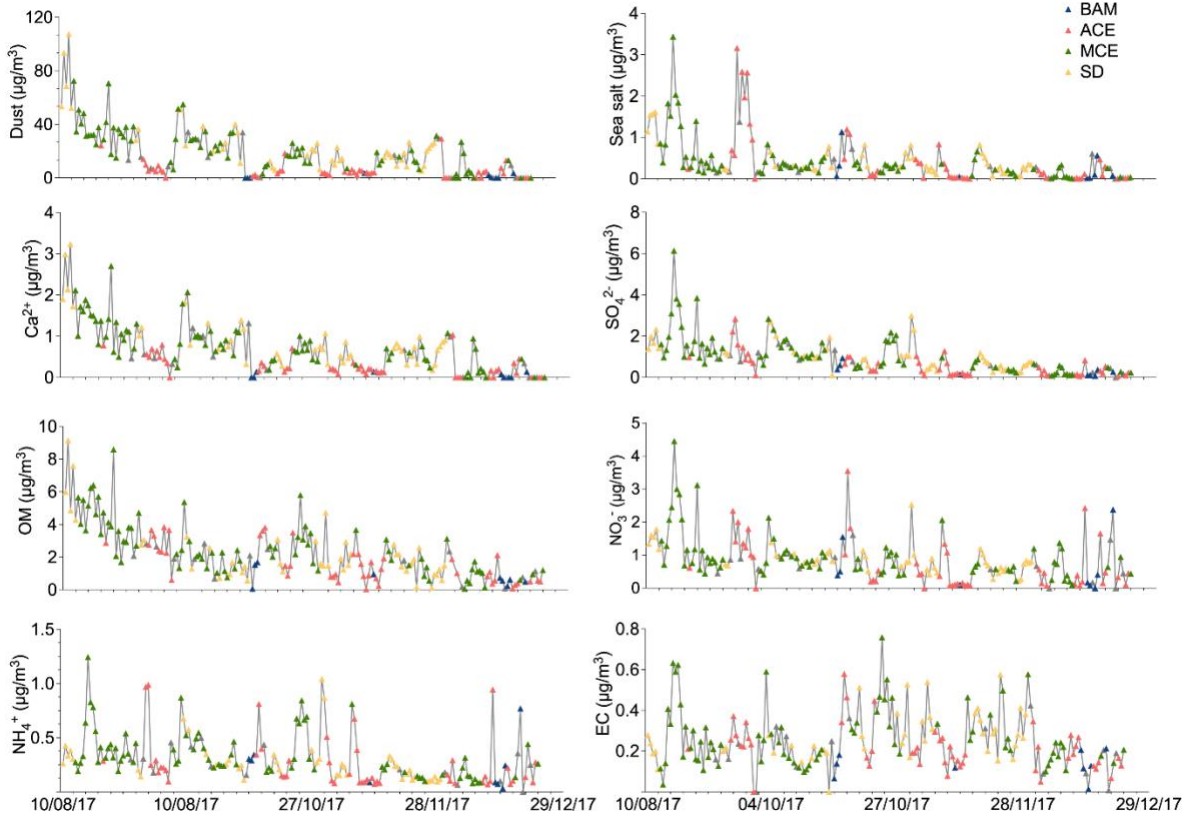

**Figure 5.** Time series of major aerosol chemical constituents in PM$_{10}$ filter samples collected from August to December 2017; The color of the symbols displayed for each sample represents a specific air mass origin: Background (blue); ACE (red); MCE (green); SD (yellow).

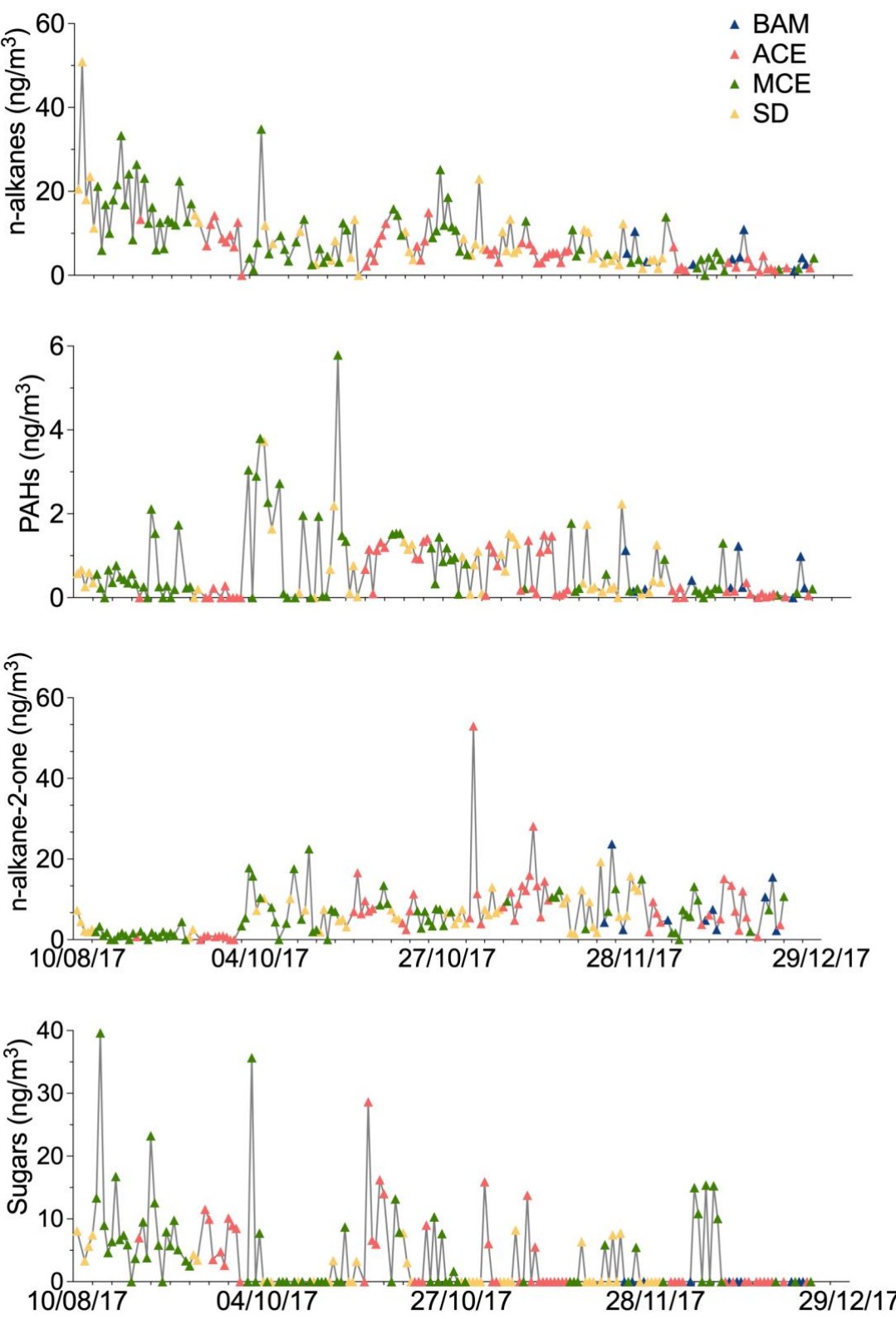


**Figure 6.** Time series of organic compounds in $PM_{10}$ filter samples collected from August to December 2017 at AMV; The color of the symbols displayed for each sample represents a specific air mass origin: Background (blue); ACE (red); MCE (green); SD (yellow).

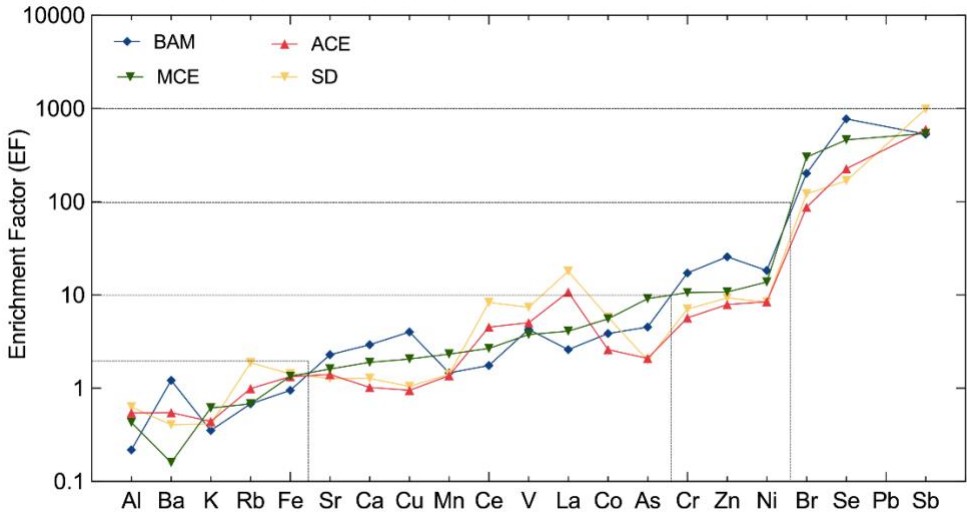


**Figure 7.** Crustal enrichment factors (EF) of aerosol $PM_{10}$ evaluated for the different trace metal elements at AMV; The averaged values are plotted according to their respective air mass origins.


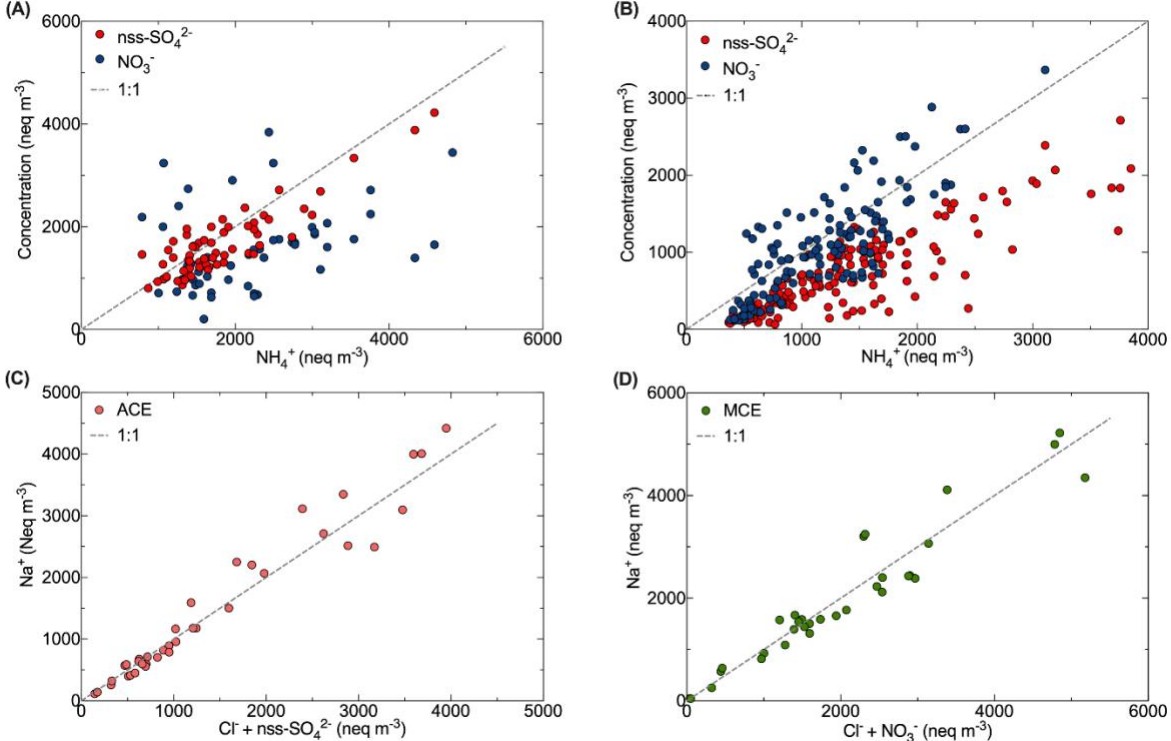

**Figure 8.** Scatter plot of (A) $NH_4^+$ with $NO_3^-$ and nss-$SO_4^{2-}$ during summer; (B) $NH_4^+$ with $NO_3^-$ and nss-$SO_4^{2-}$ during autumn-winter; (C) $Na^+$ and $Cl^- + SO_4^{2-}$ during ACE air mass; (D) $Na^+$ and $Cl^- + NO_3^-$ during

MCE air mass at the AMV site.

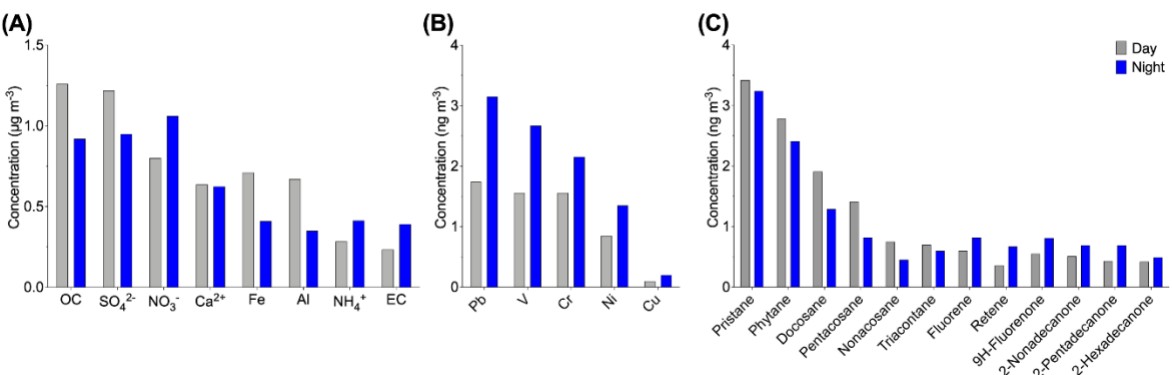

**Figure 9.** Day and night-time variation of (A) OC, EC, ionic chemical species, (B) anthropogenic metals, and (C) Organic compounds such as alkanes, PAHs and alkane-2-ones.

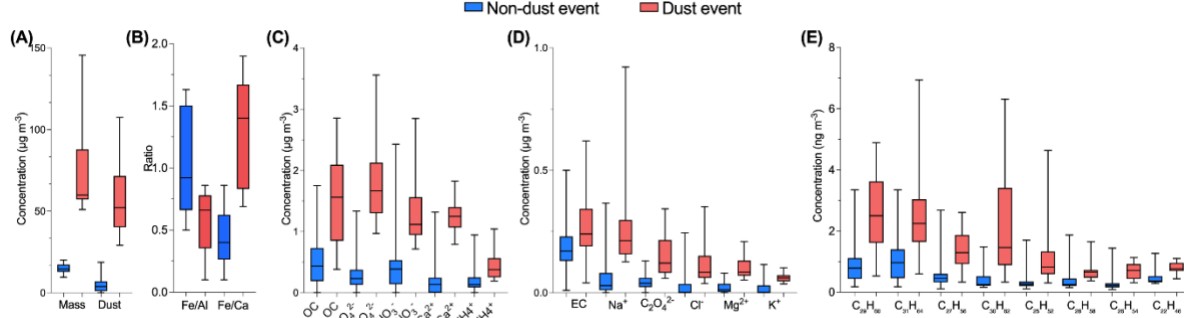


**Figure 10.** Boxplot in the period of dust and non-dust events for concentrations of **(A)** $PM_{10}$ mass and mineral dust; **(B)** Fe/Al and Fe/Ca ratio **(C)** OC and major water-soluble ions, **(D)** EC and minor water-soluble ions; and **(E)** organic compounds.


**Table 1.** Meteorological parameters over Middle-Atlas from summer 2017 to spring 2018

| Meteorological Parameter | Period | | | |
|---|---|---|---|---|
| | Summer 2017 | Autumn 2017 | Winter 2018 | Spring 2018 |
| Temperature (°C) | | | | |
| Mean | 24 | 17 | 5 | 10 |
| Min | 19 | 10 | -1 | 3 |
| Max | 26 | 21 | 7 | 13 |
| Wind speed (m s$^{-1}$) | | | | |
| Mean | 5 | 5 | 6 | 7 |
| Min | 1 | 2 | 2 | 3 |
| Max | 8 | 20 | 20 | 18 |
| Relative humidity (%) | | | | |
| Mean | 37 | 39 | 74 | 62 |
| Max | 80 | 92 | 97 | 94 |
| Rainfall (mm) | 37 | 63 | 141 | 56 |
| Wind direction (degrees) | 182 | 185 | 205 | 221 |
| Visibility (km) | 10 | 10 | 9 | 9 |
| Pressure (mbar) | 1 022 | 1 025 | 1 034 | 1 027 |


**Table 2.** Comparison of different methods for dust estimation

| Method | Mean | Max | Min | Std | Equation | Reference |
|---|---|---|---|---|---|---|
| Method 1 | 24.5 | 132.1 | 5.19 | 18.2 | $MD_1 = PM10\ mass - \left(\sum All\ detected\ element\right)$ | (Maenhaut et al., 2005)) |
| Method 2 | 19.9 | 112.3 | 0.46 | 17.8 | $MD_2 = 1.16\,(1.90\ Al + 2.15\ Si + 1.41\ Ca + 2.09\ Fe + 1.67\ Ti)$ | Maenhaut et al., 2005) |
| Method 3 | 18.9 | 104.9 | 0.20 | 18.9 | $MD_3 = 4\left(\frac{Al}{27}\right)51 + 100\left(\frac{Ca}{40}\right) + 84\left(\frac{Mg}{24}\right) + 80\left(\frac{Ti}{48}\right) + 87\left(\frac{Mn}{55}\right) + 80\left(\frac{Fe}{56}\right)$ | Minguillón et al. (2007); Nerriere et al. (2007) |
| Method 4 | 15.5 | 91.4 | 0.13 | 13.9 | $MD_4 = 2.1\ Al + 2.9\ Si + 1.4\ nss - Ca^{2+} + 1.4\ nss - Mg^{2+} + 1.43\ Fe + 1.55\ K + 1.58\ Mn$ | Cesari et al. (2012); Marenco et al. (2006); Perrino et al. (2014) |


**Table 3.** Average mass concentration of $PM_{10}$ from other high-altitude sites and urban sites in Morocco reported in the literature according to altitude.

| N° | Site | Site type | Sampling Period | Altitude (m) | $PM_{10}$ ($\mu g\ m^{-3}$) | References |
|---|---|---|---|---|---|---|
| 1 | Mt. Everest, Nepal | High altitude | Feb 2006-May 2008 | 5079 | 6 | Decesari et al., 2010 |
| 2 | Lhasa, Tibet | High altitude | Jan-Feb 2006 | 3663 | 37 | Wang et al., 2015 |
| 3 | Jungfraujoch, Switzerland | High altitude | Feb-Mar 2005 | 3580 | 3 | Cozic et al. 2008 |
| 4 | Izaña, Canary Islands | High altitude | Feb 2008-Aug 2013 | 2400 | 46 | García et al., 2017 |
| 5 | Mount Cimone, Italy | High altitude | Jun-Aug 2004 | 2165 | 16 | Marenco et al. 2006 |
| 6 | **Atlas (AMV), Morocco** | **High altitude** | **Aug-Dec 2017** | **2100** | **29** | **Present study** |
| 7 | Puy de Dome, France | High altitude | Apr 2006-Apr 2007 | 1465 | 6 | Bourcier et al. 2012 |
| 8 | Mahabaleshwar, India | High altitude | Jun 2012-May 2013 | 1348 | 37 | Leena et al., 2017 |
| 9 | Djarjeeling, India | High altitude | Jan-Dec 2005 | 2194 | 29 | Chatterjee et al. 2010 |
| 10 | Marrakech, Morocco | Urban | 2009-2012 | 465 | 55 | Inchaouh et al., 2017 |
| 11 | Meknes, Morocco | Urban | Mar 2007-Apr 2008 | 546 | 47 | Tahri et al., 2017 |
| 12 | Tetouan, Morocco | Urban | May 2011-Apr 2012 | 105 | 31 | Benchrif et al., 2018 |
| 13 | Kenitra, Morocco | Urban | Feb 2007-Feb 2008 | 26 | 110 | Tahri et al., 2013 |


**Table 4** Concentrations of main aerosol chemical species in $PM_{10}$ according to each air mass ($\mu g\ m^{-3}$) at AMV; The organic composition is given in ng $m^{-3}$; The number of samples is written in parentheses.

| Aerosol components | Air mass | | | |
|---|---|---|---|---|
| | BAM (n=10) | ACE (n=51) | MCE (n=71) | SD (n=43) |
| Mass load | $10.9 \pm 0.9$ | $20.4 \pm 6.3$ | $33.8 \pm 14.5$ | $37.9 \pm 25.3$ |
| Dust | $5.5 \pm 3.5$ | $13.3 \pm 5.2$ | $19.9 \pm 11.9$ | $29.1 \pm 22.6$ |
| Sea salt | $0.05 \pm 0.06$ | $0.3 \pm 0.5$ | $0.3 \pm 0.4$ | $0.2 \pm 0.2$ |
| OM | $1.0 \pm 0.6$ | $1.4 \pm 1.1$ | $2.7 \pm 1.7$ | $2.3 \pm 1.8$ |
| EC | $0.2 \pm 0.1$ | $0.2 \pm 0.1$ | $0.2 \pm 0.1$ | $0.2 \pm 0.1$ |
| POC | $0.1 \pm 0.06$ | $0.2 \pm 0.3$ | $0.3 \pm 0.3$ | $0.2 \pm 0.2$ |
| SOC | $0.3 \pm 0.2$ | $0.4 \pm 0.4$ | $1.0 \pm 0.8$ | $0.9 \pm 0.8$ |
| $NO_3^-$ | $0.5 \pm 0.6$ | $0.6 \pm 0.7$ | $1.0 \pm 0.7$ | $0.9 \pm 0.4$ |
| nss-$SO_4^{2-}$ | $0.2 \pm 0.2$ | $0.5 \pm 0.5$ | $1.2 \pm 0.9$ | $1.0 \pm 0.6$ |
| $NH_4^+$ | $0.2 \pm 0.2$ | $0.2 \pm 0.2$ | $0.3 \pm 0.2$ | $0.2 \pm 0.1$ |
| $Ca_2^+$ | $0.2 \pm 0.2$ | $0.2 \pm 0.2$ | $0.8 \pm 0.5$ | $0.9 \pm 0.6$ |
| Alkanes | $4.9 \pm 3.2$ | $5.6 \pm 3.7$ | $10.5 \pm 7.7$ | $9.2 \pm 8.6$ |
| PAHs | $0.4 \pm 0.5$ | $0.4 \pm 0.4$ | $0.9 \pm 2.1$ | $0.7 \pm 0.7$ |
| Alkan-2-ones | $7.8 \pm 6.9$ | $5.9 \pm 5.5$ | $5.5 \pm 5.0$ | $6.6 \pm 4.1$ |
| Sugars | - | $3.7 \pm 6.0$ | $5.2 \pm 7.7$ | $1.8 \pm 2.9$ |
| Oxalate | $44 \pm 26$ | $73 \pm 58$ | $129 \pm 58$ | $107 \pm 63$ |
| pH | $5.6 \pm 0.2$ | $6.0 \pm 0.4$ | $6.5 \pm 0.4$ | $6.5 \pm 0.4$ |
| OC/EC | $2.2 \pm 1.1$ | $3.3 \pm 1.9$ | $6.3 \pm 7.5$ | $4.6 \pm 4.6$ |
| CPI | $3.3 \pm 0.8$ | $3.5 \pm 2.4$ | $3.9 \pm 1.9$ | $4.0 \pm 3.1$ |

**Table 5.** Concentrations of main aerosol chemical species in $PM_{10}$ (ng m$^{-3}$) at AMV compared to other high altitude mountain stations. Data are reported in the format average (mean ± standard deviation) and NA: not available. Notice that the concentrations of $PM_{10}$ mass are given in µg m$^{-3}$. [a] Present study; [b] Bourcier et al. 2012; [c] Chatterjee et al. 2010; [d] Marenco et al. 2006; [e] Decesari et al., 2010.

| Elements | Mt. Atlas, Morocco[a] | Mt. Puy de Dome, France[b] | Mt. Himalaya, India[c] | Mt. Cimone, Italy[d] | Mt. Everest, Nepal[e] |
|---|---|---|---|---|---|
| Altitude (m a.s.l) | 2100 m | 1465 m | 2194 m | 2165 m | 5079 m |
| Samples | 190 | NA | 111 | 57 | 99 |
| Period | Aug-Dec 2017 | Apr 2006-Apr 2007 | Jan-Dec 2005 | Jun-Aug 2004 | Apr 2006 - May 2008 |
| Mass load | 29.1 ± 17.3 | 5.6 ± 4.6 | 29.5 ± 20.8 | 16.1 ± 9.8 | 5.6 ± 4.6 |
| OC | 1069 ± 818 | NA | NA | NA | 800 ± 637 |
| EC | 247 ± 134 | NA | NA | NA | 115 ± 132 |
| $Na^+$ | 186 ± 231 | NA | 2200 ± 2000 | NA | 24.2 ± 22.5 |
| $K^+$ | 42 ± 35 | NA | 310 ± 210 | NA | 34 ± 32 |
| $Ca^{2+}$ | 649 ± 579 | 15.5 ± 10.2 | 130 ± 10 | 360 ± 550 | 138 ± 90 |
| $Mg^{2+}$ | 60 ± 50 | NA | 120 ± 60 | NA | 19.3 ± 7.2 |
| $Cl^-$ | 80 ± 133 | NA | 2350 ± 1500 | 82 ± 98 | 22 ± 46 |
| $NH_4^+$ | 298 ± 220 | 297 ± 276 | 50 ± 40 | 1400 ± 800 | 175 ± 183 |
| $NO_3^-$ | 859 ± 687 | 510 ± 980 | 950 ± 200 | 840 ± 770 | 170 ± 223 |
| $SO_4^{2-}$ | 941 ± 848 | 1380 ± 1160 | 3500 ± 2100 | 3500 ±2000 | 394 ± 329 |
| Al | 443 ± 830 | NA | NA | 300 ± 460 | 740 |
| Fe | 486 ± 728 | NA | NA | 260 ± 440 | NA |
| Ti | 37 ± 45 | NA | NA | 30 ± 50 | NA |
| V | 3.5 ± 12.2 | NA | NA | 3.1 ± 1.5 | NA |
| K | 174 ± 156 | NA | NA | 160 ± 210 | NA |
| Cr | 4.3 ± 5.2 | NA | NA | NA | NA |
| Ni | 2.4 ± 3.1 | NA | NA | 1.4 ± 0.5 | NA |
| Cu | 1.2 ± 3.1 | NA | NA | 2.9 ± 3.1 | NA |
| Zn | 8.6 ± 6.2 | NA | NA | 9.9 ± 6.6 | NA |
| Pb | 4.8 ± 4.5 | NA | NA | 3.9 ± 2.4 | NA |
| Mn | 12.4 ± 39.3 | NA | NA | 6.2 ± 7.0 | NA |
