# Peer review of "First insights into Northern Africa high-altitude background aerosol chemical composition and source influences"

_Atmospheric Chemistry and Physics, 2021_

## Author Response (AR1)

We thank referee #1 R Subramanian for the many constructive comments and suggestions which helped to improve the manuscript. In order to improve readability, we numbered each reviewer comment and its corresponding response in the style R1-C1 for reviewer 1 comment 1and R1-A1 for reviewer 1 answer to comment 1, respectively. Comments by the reviewer are given in black, our response to the comments are shown in red, and modified text in the revised manuscript are given in green. Supporting information (SI) has been added. In the following, we provide a point-by-point response to all comments.

**RC1: 'Important submission on an understudied region', R Subramanian, 05 Jul 2021**

**R1-C1:** Deabji et al. present initial (five-month) PM10 and chemical composition results from a new monitoring station at a remote elevated site in Northern Africa. Given the lack of data from this region, this is an important contribution that will lead to greater understanding of aerosol sources in Northern Africa. Unfortunately, the manuscript as presented is hard to read and could benefit from restructuring/reorganization. Some of the conditions and comparisons seem arbitrary.

**R1-A1:** The manuscript has been copy-edited and large parts of the manuscript have been re-written and thoroughly improved. Figures have been improved. Grammatic errors have been corrected and the language has been edited to be more concise throughout. The references have been edited to avoid errors in the citations. The structure of the manuscript has been improved. The classification criteria have been reconsidered, therefore some of the results have been modified. Some sections have been combined to improve readability, comparisons, and avoid repetition. A point-by-point description of the changes made to the manuscript is given in the following.

**R1-C2:** For example, they report PM10 concentrations during "background" periods were up to 20 µg/m3 - but that is because the authors set 20 µg/m3 as the upper limit defining "background" periods.

**R1-A2:** The choice of the background definition was motivated by different studies such as Puxbaum et al., 2004; Vardoulakis and Kassomenos, 2008; Karaca et al., 2009; Harrison et al., 2004; Escudero et al., 2006 who suggested that the reference could be at 20 $\mu$g m$^{-3}$. However, as recommended by the reviewer the lowest fifth (5$^{th}$) percentile of PM$_{10}$ mass concentrations has now been considered as the baseline threshold. This can be easily seen in the new PM$_{10}$ variation Figure (Fig. 3A). Based on these new criteria, the PM$_{10}$< 12 $\mu$g m$^{-3}$ was considered as background. The number of samples that med this condition was 10 and the mean concentration was about 10.9 $\mu$g m$^{-3}$. The frequency and the probability density function of mass concentrations and wind speed were calculated for background samples (Fig. S3). The thresholds chosen are in line with the results obtained from statistical calculations, as highlighted with the blue bars in the frequency and probability density function Figures below.

[Figure]

**Figure S3.** Frequency distribution and corresponding probability density function of $PM_{10}$ mass and wind speed during the sampling period.

The following text has been added on the manuscript on lines 318-323:

*"To establish a reference baseline and evaluate the background conditions at the site, the lower 5th percentile of the $PM_{10}$ concentrations ($PM_{10} < 12\ \mu g\ m^{-3}$) was found to be representative of remote background aerosol conditions. The $PM_{10}$ frequency and probability density function as shown in Fig. S3 confirmed this observation. The samples within this PM concentration range had similar air mass trajectories and typical meteorological conditions with low wind speeds $< 3\ m\ s^{-1}$. The air masses typically traveled in the free troposphere at about 1000 m above sea level, crossing the North-Atlantic Ocean before arriving at the site within the past 96 h."*

**R1-C3:** They also excluded local pollution from "background" and then present results saying these were dominated by local dust – this again is a result of excluding other local sources.

**R1-A3:** The former criterion has been revised and new chemical composition of these samples evaluated. The current criteria also consider both local and regional sources during background conditions especially from anthropogenic activities such as local road dust resuspension from cars assessing the site. A corresponding sentence has been added to the manuscript in line 324 and now reads:

*"These conditions were, however, not free from local and regional pollution from point sources such as dust resuspension from the cars accessing the site."*

**R1-C4:** A better approach might be to pick the lowest fifth (5th) or tenth (10th) percentile of PM$_{10}$ mass concentrations and describe that as background condition.

**R1-A4:** We thank the reviewer for the suggestion which is crucial for the selection of background samples. As recommended by the reviewer the lowest fifth (5th) percentile of PM$_{10}$ mass concentrations has now been considered as the baseline threshold for the background conditions. This can be easily seen in the new PM$_{10}$ variation figure, as shown below in the updated Fig. 3a. Based on these new criteria, the maximum PM$_{10}$ during these conditions was 12 µg m$^{-3}$. The number of samples that met this condition was 10 and the mean concentration was about 10.9 µg m$^{-3}$. A corresponding line has been added on the manuscript on lines 326-329:

*"Consequently, the average background PM$_{10}$ mass concentration at the Middle-Atlas was 10.9 µg m$^{-3}$, which was found to be stable and representative of periods of little external influence. In comparison, Benchrif et al. (2018) reported background PM$_{10}$ values for Northern Morocco with an average of 12.2 µg m$^{-3}$, which is very similar to the concentrations determined in this study, 10.9 µg m$^{-3}$."*

[Figure]

**Figure 3.** (A) Time series of daily PM$_{10}$ mass

**R1-C5:** Similarly, the comparisons with Delhi and Kathmandu seem out of place, as pollution in urban centers especially these locations is very complex and unlike AM5. Since the focus of this manuscript is on AM5, even comparing Moroccan cities (from other published literature, not this work) to Delhi is out of place in this manuscript.

**R1-A5:** The comparison section has been restructured and rewritten and Table 3 has been modified. The choice of stations was redesigned with a focus on remote background stations. The GAW network database was used as a reference for site selection. Sites like Jungfraujoch, Mt. Everest, Mt. Cimone, and Puy de Dôme were introduced to optimize and make the comparison more relevant. The choice of urban stations was also questioned, therefore only urban stations located in Morocco are now included. The update of Table 3 is as follows:

[revised manuscript text omitted]

*In lines 517-519:* *"The concentrations of sulfate (0.9 ±0.8 $\mu g$ $m^{-3}$) over AM5 were comparable with those at Puy de Dôme (1.3 ± 1.1 $\mu g$ $m^{-3}$), but were almost 4-5 times lower than all the other hilly stations except Mt. Everest (Decesari et al., 2010)."*

**R1-C6:** I also had a hard time following the paper, and it could benefit from a more organized structure. For example, in the site intercomparison above, the authors compare AM5 with Izana (Moroccan coastal site), then with India/Tibet/Kathmandu, then back to Cape Verde (off the coast of North Africa), then to an unnamed background site in the Mediterranean. This discussion would be more cohesive if AM5 is compared first with all remote sites, then (if at all) with urban sites rather than switching back and forth.

**R1-A6:** The intercomparison section was rewritten and reworded following the reviewer's recommendations. Starting first with a comparison of station AM5 with remote high-altitude sites, then a comparison with Moroccan urban sites was introduced in the following paragraph. The intercomparison section has been added to the revised manuscript on lines 331-340 and now reads:

*"The mean concentration recorded at AM5 from this study ($29.2 \pm 17.3 \, \mu g \, m^{-3}$) agreed well with the $PM_{10}$ concentration of other remote high-altitude sites, such as Darjeeling in Northeastern Himalayas ($29 \, \mu g \, m^{-3}$; Chatterjee et al. 2010), Lhasa in Tibet ($37 \, \mu g \, m^{-3}$; Wang et al., 2015), and Mahabaleshwar in India ($37 \, \mu g \, m^{-3}$; Leena et al., 2017) as presented in Table 3. Other high-altitude stations, such as Izaña, in Canary Islands ($46 \, \mu g \, m^{-3}$) showed much higher $PM_{10}$, most likely due to the exposure to strong Saharan dust events (García et al., 2017). In contrast, the $PM_{10}$ concentrations at AM5 were considerably higher than the $PM_{10}$ levels recorded in European and Asian high-altitude sites. For example, the average $PM_{10}$ mean value recorded in this study was about twice that of Mount Cimone, Italy ($16 \, \mu g \, m^{-3}$; Marenco et al., 2006) and factor 6 greater than the $PM_{10}$ in Everest Mountain (Decesari et al., 2010) and in Puy de Dôme, France ($6 \, \mu g \, m^{-3}$; Bourcier et al. 2012), and was approximately 10 times greater than the average level in Jungfraujoch ($3 \, \mu g \, m^{-3}$; Cozic et al., 2008)."*

**R1-C7:** when discussing daily and monthly variations, they first discuss August/summer. In the next paragraph, they discuss Oct-Nov-Dec, but midway again discuss August and Saharan influence, then again present winter (Nov-Dec?) results.

**R1-A7:** The daily and monthly discussion has been restructured in a uniform and consistent paragraph. The variation during summer has been moved to the previous paragraph followed by the discussion on the autumn and winter. The corresponding sentences on lines 234-335 and now read:

*"During August, high $PM_{10}$ concentration were mostly related to high wind speeds from southeast wind direction. For example, $PM_{10}$ mass concentration often exceeded $50 \, \mu g \, m^{-3}$ and sometimes even reached up to $145 \, \mu g \, m^{-3}$ during August, when the wind speed was stronger than $9 \, m \, s^{-1}$."*

**R1-C8:** Even the winter discussion is inconsistent, as they first say a change in wind direction leads to lower $PM_{10}$ (which seems unlikely since the populated centers are to the west and wind speeds are high), but then attribute the lower PM to increased precipitation (which is more logical).

R1-A8: We thank the reviewer for this insightful comment. The winter discussion has been modified and the precipitation argument reworded to make it more consistent and uniform. The drop in PM is certainly due to precipitation. The corresponding sentences in lines 347-352 now read:

*"A sharp fall in PM concentrations was noticed in November (22.8 ± 7.9 µg m$^{-3}$) and December (15.9 ± 5.6 µg m$^{-3}$). Overall, PM$_{10}$ concentration decreased from the summer to winter by 32%. This trend is most likely due to the increased amount of precipitation (peaks of 852mm) during fall and winter, which can lead to the wash-out effect of aerosol and its components (Holst et al. 2008)."*

**R1-C9:** Skimming the rest of the paper showed continued repetition and disorganized presentation of results - for example, line 822 has the "first insight into aerosol chemical composition..." midway through a paragraph, and then next paragraph also starts with "first high altitude aerosol characterization study..."

**R1-A9:** The repetitions have been eliminated and the manuscript has been edited. All the copy editing modifications are listed at the end of the response.

**R1-C10:** I also don't know why sulfate concentration of 1.4 µg/m$^3$ is presented as high when average concentrations at AM5 are ~30 µg/m$^3$ - is that because that sulfate level is high relative to average sulfate at the site? It might help to separate these two results, then.

R1-A10: That sentence was meant to compare the sulfate concentrations with those during background conditions, Therefore, the two results were separated as suggested. In the first instance, the sulfate concentration of MCE is high (1.2 ± 0.9 µg m$^{-3}$) compared to the average sulfate concentration during background conditions 0.2 ± 0.2 µg m$^{-3}$ at the AM5 site. The corresponding sentences have been added as follows and now read:

*In lines 500-501: "In average, the sulfate concentration for MCE (1.2 ± 0.9 µg m$^{-3}$) was about 5 times higher than background sulfate concentrations (0.2 ± 0.2 µg m$^{-3}$)."*

*In lines 517-519: "The concentrations of sulfate (0.9 ± 0.8 µg m$^{-3}$) over AM5 were comparable with those at Puy de Dôme (1.3 ± 1.1 µg m$^{-3}$), but were almost 4-5 times lower than all the other hilly stations except Mt. Everest (Decesari et al., 2010)."*

**R1-C11:** The next few sections were lengthy, somewhat repetitive descriptions of chemical composition results. The manuscript may be more readable if the monthly variation and chemical composition results are replaced and reorganized by air mass as winds from different directions seem to hit the site each month.

**R1-A11:** The manuscript has been restructured by introducing air mass classification before the description of the chemical composition. As a result, the new structure allows the discussion of the results of the chemical composition of the complete period using the interpretation of the influence of the air masses introduced previously, to avoid any repetitions. The modified structure now reads as follow:

*3.1 Variation of PM10 mass*

*3.2 Air mass origins*

*3.3 Characterization of aerosol chemical composition for the complete measurement period*

*3.4 Inter-relationship between aerosol components*

*3.5 Day and night-time variation*

*3.6 Differences in chemical composition between dust and non-dust events*

The section of PM$_{10}$ mass variation according to month and chemical composition has been reorganized by introducing the influence of air masses and winds coming from different directions. Figure 3 was modified to discuss the variation of PM$_{10}$ concentration according to the change of wind speed and direction of each month. A new Fig .3 is shown below.

[Figure]

**Figure 3.** (A, top right) Time series of daily PM$_{10}$ mass; (B, top left) Box plot of monthly averages of PM$_{10}$ mass; (C, bottom) Pollution rose of PM$_{10}$ mass; The presented data were separated according to each season into summer (Aug), Fall (Sep-Nov), and winter (Dec).

**R1-C12:** Maybe the authors could shorten and reorganize the paper as NAO+MCE (as they are similar); ACE; SD; and local remote/background - these are perhaps more interesting for atmospheric research and future campaign planning than monthly data.

**R1-A12:** We thank the reviewer for his suggestion to reorganize the air mass classification section. Accordingly, the manuscript has been organized according to the reviewer's suggestion, the NOA and ACE air masses have been grouped, and Background Air masses; (BAM) have been introduced in the classification. Figure 4 has been modified as follows:

[Figure]

**Figure 4.** Typical 96h air mass back trajectory performed for AM5 during routine samples periods; aerosol type and $PM_{10}$ mass concentration are given in parentheses: (a) 18 December 2017: air mass from the North Atlantic Ocean considered to be representative of background conditions (Background, $m$=10.9 µg m$^{-3}$); (b) 10 October 2017: air mass from Europe crossing coastline of North Morocco (Atlantic Coast Europe, $m$=44.1 µg m$^{-3}$); (c) 2 November 2017: slightly polluted air mass from North East crossing Mediterranean Sea (Mediterranean Coast Europe, $m$=26.4 µg m$^{-3}$) ; (d) 13 August 2017: dust loaded air mass coming from Sahara desert (Saharan Dust, $m$=94.1 µg m$^{-3}$).

The paragraph describing the results of updated air mass classification in section air mass origins (3.2) has been modified. The corresponding sentences have been modified in lines 347-357 as follow and now read:

*"The calculation of back trajectories using the Hysplit model allowed the identification of several remote sources of $PM_{10}$ at the station. Four main air masses categories were identified as shown in Fig. 4. i) Air masses that spent the last 96h over the Atlantic Ocean at high-altitude (1000 m.asl), representative of typical background air mass (BAM) conditions, which influenced about 5.3% of all samples; ii) Air masses originating from the Atlantic and crossing over the Coast of Europe (ACE), especially Spain and over Moroccan industrial cities located at the North Atlantic coast, influencing about accounts 26.8% of all samples; iii) Air masses from Europe crossing over the Mediterranean Coast and Europe (MCE) as well as over North Moroccan cities, as shown in Fig. 4c and influencing about 37.4% of all samples and iv) Air masses originating from Southern and/or Eastern Sahara crossing the desert (SD), at different altitudes before arriving at the AM5 and influencing about 22.6% of all samples. The remaining back trajectories represent mixing scenarios including 7.9% of all samples that could not be assigned to the four major classes mentioned above."*

Consequently, the chemical composition corresponding to each air mass has been modified, as presented in Table 4. The concentrations of the compounds have been modified according to the new classification.

**Table 4.** $PM_{10}$ concentrations ($\mu g\ m^{-3}$) of main aerosol chemical species according to the air mass influence at AM5; The organic composition is given in ng $m^{-3}$; The number of samples is written in parentheses.

| Aerosol components | Air mass | | | |
| --- | --- | --- | --- | --- |
| | BAM (n=10) | ACE (n=51) | MCE (n=71) | SD (n=43) |
| Mass load | $10.9 \pm 0.9$ | $20.4 \pm 6.3$ | $33.8 \pm 14.5$ | $37.9 \pm 25.3$ |
| Dust | $5.5 \pm 3.5$ | $13.3 \pm 5.2$ | $19.9 \pm 11.9$ | $29.1 \pm 22.6$ |
| Sea salt | $0.05 \pm 0.06$ | $0.3 \pm 0.5$ | $0.3 \pm 0.4$ | $0.2 \pm 0.2$ |
| OM | $1.0 \pm 0.6$ | $1.4 \pm 1.1$ | $2.7 \pm 1.7$ | $2.3 \pm 1.8$ |
| EC | $0.2 \pm 0.1$ | $0.2 \pm 0.1$ | $0.2 \pm 0.1$ | $0.2 \pm 0.1$ |
| POC | $0.1 \pm 0.06$ | $0.2 \pm 0.3$ | $0.3 \pm 0.3$ | $0.2 \pm 0.2$ |
| SOC | $0.3 \pm 0.2$ | $0.4 \pm 0.4$ | $1.0 \pm 0.8$ | $0.9 \pm 0.8$ |
| $NO_3^-$ | $0.5 \pm 0.6$ | $0.6 \pm 0.7$ | $1.0 \pm 0.7$ | $0.9 \pm 0.4$ |
| $nss\text{-}SO_4^{2-}$ | $0.2 \pm 0.2$ | $0.5 \pm 0.5$ | $1.2 \pm 0.9$ | $1.0 \pm 0.6$ |
| $NH_4^+$ | $0.2 \pm 0.2$ | $0.2 \pm 0.2$ | $0.3 \pm 0.2$ | $0.2 \pm 0.1$ |
| $Ca_2^+$ | $0.2 \pm 0.2$ | $0.2 \pm 0.2$ | $0.8 \pm 0.5$ | $0.9 \pm 0.6$ |
| Alkanes | $4.9 \pm 3.2$ | $5.6 \pm 3.7$ | $10.5 \pm 7.7$ | $9.2 \pm 8.6$ |
| PAHs | $0.4 \pm 0.5$ | $0.4 \pm 0.4$ | $0.9 \pm 2.1$ | $0.7 \pm 0.7$ |
| Alkan-2-ones | $7.8 \pm 6.9$ | $5.9 \pm 5.5$ | $5.5 \pm 5.0$ | $6.6 \pm 4.1$ |
| Sugars | - | $3.7 \pm 6.0$ | $5.2 \pm 7.7$ | $1.8 \pm 2.9$ |
| Oxalate | $44 \pm 26$ | $73 \pm 58$ | $129 \pm 58$ | $107 \pm 63$ |
| pH | $5.6 \pm 0.2$ | $6.0 \pm 0.4$ | $6.5 \pm 0.4$ | $6.5 \pm 0.4$ |
| OC/EC | $2.2 \pm 1.1$ | $3.3 \pm 1.9$ | $6.3 \pm 7.5$ | $4.6 \pm 4.6$ |
| CPI | $3.3 \pm 0.8$ | $3.5 \pm 2.4$ | $3.9 \pm 1.9$ | $4.0 \pm 3.1$ |

The comparison between the chemical composition with respect to the air masses (Table 4) has been included in the discussion paragraphs in section (3.3). The corresponding sentences have been added as follow:

*In lines 383-389: "Mineral dust was found to be more than 7 times higher (37.9 ± 25.3 µg m-3) during dust events (SD), in comparison to remote background conditions (BAM) with an average concentration of 5.5 ± 3.5 µg m$^{-3}$, as observed in Table 4. Other less intense Saharan dust storms occurred during the summer season between the 21$^{st}$ and 24$^{th}$ of August with a similar high dust concentration that was 5 times higher than dust background concentrations. The presence of mineral dust is relatively low but still significant for air masses other than SD, such as during the ACE (13.3 ± 5.2 µg m$^{-3}$) and MCE (19.9 ± 11.9 µg m$^{-3}$) air mass influence, as shown in Table 4."*

*In lines 444-445: "The highest OC/EC ratio (6.3 ± 7.5) was observed for MCE air masses, while the lowest ratio was recorded for BAM at about 2.3 ± 1.1, as shown in Table 4."*

*In lines 462-463: "However, no significant difference was noticed in sea salt concentrations found in the ACE (0.5 ± 0.7 µg m$^{-3}$) and MCE samples (0.5 ± 0.5 µg m$^{-3}$), as shown in Table 4."*

*In lines 494-496: "On average, the influence of long-range transport during the ACE and MCE air masses for sulfate (2.8 µg m$^{-3}$) and nitrate (2.3 µg m$^{-3}$) were similar. However, the contribution of ammonium (1.7 µg m-3) to particulate matter was particularly higher for MCE air mass."*

*In lines 550-552: "Table 4 presents the CPI values calculated according to each air mass. The average CPI value was 3.8 ± 2.4 and ranged from 0.7 to 18.6. However, high CPI (>>1) was observed for all air masses, which indicates that the alkanes originated from plants waxes, as presented in Table 4 (Kavouras, 2002)."*

*In lines 570-575: "The highest amount of PAH was detected during October, approximately 5.7 ng m$^{-3}$ due to long-range transport of MCE air masses, as shown in Figure 6. The minimum concentration was observed during winter, of about 0.05 ng m$^{-3}$. The average background concentration of PAHs was 0.4 ± 0.5 µg m$^{-3}$, which was low in comparison to other organic compounds likely because of high evaporation on warm days (Cincinelli et al., 2007). During MCE air mass, the PAH concentrations increased by 52% compared to the BAM concentration, as shown in Table 4."*

*In lines 630-641: "The average sugar concentrations during these air mass influences were ACE (3.7 ± 6.0 ng m$^{-3}$) and MCE (5.2 ± 7.7 ng m$^{-3}$), as listed in Table 4. In contrast, sugar compounds were relatively low in SD (1.8 ± 2.9 ng m$^{-3}$) air masses and were not found in the background PM$_{10}$ conditions. Levoglucosan which is considered as a good tracer of biomass burning emissions in aerosol particulate matter, was particularly higher for ACE (2.0 ng m$^{-3}$) and MCE (1.6 ng m$^{-3}$) air mass influence, as displayed in Fig. S8. (Bauer et al., 2008). Arabitol shows a similar concentration for MCE and ACE with a mean of 1.0 ng m$^{-3}$ suggesting that particles were loaded with primary biological aerosols such as pollen, fungal spores, vegetative debris, viruses, and bacteria from the marine coast (Fu et al., 2012). Glucose remained relatively high during MCE air mass influence in comparison to other air masses influence. During SD air mass influence, the concentration of arabitol was extremely low with a concentration less than 0.08 ng m$^{-3}$. However, glucose showed a higher concentration of about 0.7 ng m$^{-3}$ but remains 3 times lower than MCE concentrations. This indicates that the sugars were most likely originated from marine air masses."*

The background shading type has been modified to improve visibility. Thus, each sample is represented by a distinguishing color symbol for each specific air mass. At the same time, Figure 6, which shows the temporal variation of the organics, has been modified in the same way. The new Figure 5 and Figure 6 are shown below:

[Figure]

**Figure 5.** Time series of major aerosol chemical constituents in PM$_{10}$ filter samples collected from August to December 2017; The color of the symbols displayed for each sample represents a specific air mass origin: Background (blue); ACE (red); MCE (green); SD (yellow).

Consequently, the description of Figure 5 has been modified in section 3.3 according to the new air mass classification, so the modified paraphrases are listed as follows:

[revised manuscript text omitted]

**Further comments**

After restructuring the manuscript to improve readability and logical flow (and some light copy-editing*) the paper can be re-reviewed and considered suitable for publication.*Some examples:

**R1-C13:** Lines 160-161: please use either km/h or m/s, but not both especially when comparing two values.

**R1-A13:** The unit *"m s$^{-1}$"* was used to describe the wind speed and has been used to replace km h$^{-1}$ throughout the whole manuscript. The sentences can be read as follow:

*Linies 149-150: "The average wind speed at AM5 was about 5.8 m s$^{-1}$ but reached a maximum of 19.7 m s$^{-1}$ due to turbulence in the mountain region, especially during winter."*

*Linie 150: "Over the summer, the minimum wind speed was about 1.6 m s$^{-1}$, and the relative humidity (RH) was low."*

*Linies 163-164: "During summer, southeast winds have a higher frequency but a lower average speed of about 5.8 m s$^{-1}$."*

*Lines 165-166: "During this period, there is a strong occurrence of westerly winds which are often characterized by high wind speeds (stiff breeze) of up to 20 m s$^{-1}$."*

*Lines 299-300: "For example, PM$_{10}$ mass concentration often exceeded 50 µg m$^{-3}$ and sometimes even reached up to 145 µg m$^{-3}$ during August, when the wind speed was stronger than 9 m s$^{-1}$."*

*Lines 306-307: "High PM$_{10}$ concentrations were observed during strong westerly winds of up to > 7 m s$^{-1}$."*

*Lines 320-321: "The samples within this PM concentration range were had similar air mass trajectories and typical meteorological conditions with low wind speeds < 3 m s$^{-1}$."*

*Lines 336-438: "Low concentrations were observed during days with low wind speeds (< 2 m s$^{-1}$), low Saharan dust air mass inflow, and after precipitation events, which typically occurred in the fall and winter."*

*Lines 457-459: "The sea salt concentration was high when wind speed exceeded 6 m s$^{-1}$, indicating that sea salt is strongly dependent on meteorological conditions and air mass sources."*

*Lines 800-801: "High speeds were recorded during the night, up to 17.5 m s$^{-1}$, mostly associated with marine air masses."*

**R1-C14:** It is unclear why low wind speeds indicate Saharan dust reaches AM5 via this path.

**R1-A14:** The change in wind direction in addition to the wind speed is responsible for the transport of the Saharan dust. Nevertheless, the low wind speed from the south recorded during summer is due to the High Atlas Mountains which protect the site and act as a barrier to block the transport of Saharan mineral dust. By contrast, the marine air masses coming from the west are characterized by strong winds and can reach maximum speeds of 20 m s$^{-1}$. As result, the sentences have been corrected in section 3.3.1 in lines 368-375, and now read:

*"The influence of the Saharan dust on the Middle Atlas region remains relatively dependent on meteorological conditions. Firstly, the direction and speed of the wind, as the typical Saharan dust events were observed during high wind speed periods from south and southeast. Secondly, their progression depends mainly on favorable weather conditions for transport, the difference in temperature between day and night, humidity, and especially the scarcity of rainfall. Thirdly, the High-Atlas mountains situated at 4000 m of altitude acts as a barrier to*

*Saharan dust transport which forces the winds to deviate from their path. All these factors influence the transport of large particles from the Sahara to the Middle-Atlas during the different seasons."*

**R1-C15:** Also the authors say summer is dominated by southwestern winds, but only describe westerly and southeasterly winds.

**R1-A15:** The polar rose of $PM_{10}$ in Figure 3c has been modified accordingly for each month to discuss the variation of $PM_{10}$ concentration with respect to wind speed and direction. A description of southwestern winds has been added in lines 305-307, and now reads:

*"Furthermore, high concentrations up to 40-50 µg m$^{-3}$ were observed as well with westerly winds, especially during northwest and southwest winds. The highest $PM_{10}$ concentrations was observed during strong westerly winds of up to > 7 m s$^{-1}$."*

**R1-C16:** The last sentence of this section then says summer is dominated by southern winds, so this section is also confusingly structured/worded.

**R1-A16:** The last sentence of the section has been deleted and the paragraph has been restructured and rewritten, and the lines 298-316 now read:

*"During August, the high $PM_{10}$ concentrations were mostly related to high wind speeds from the southeast. For example, $PM_{10}$ mass concentration often exceeded 50 µg m$^{-3}$ and sometimes even reached up to 145 µg m$^{-3}$ during August, when the wind speed was stronger than 9 m s$^{-1}$. The high $PM_{10}$ concentration recorded was due to the influence of Saharan dust events during periods of air mass influence from the southern sector located in the southeast of the AM5 station. The Middle Atlas region is marked by particular meteorological conditions during the summer with low humidity and often low precipitation (avg. 37 mm), as shown in Table 1. These hot and arid conditions are known to favor the transport of dust particles from the Saharan desert to the Atlas Mountains (Rodríguez et al., 2011). Furthermore, high concentrations of up to 40-50 µg m$^{-3}$ were observed as well with westerly winds, especially during northwest and southwest winds. High $PM_{10}$ concentrations were observed during strong westerly winds of up to > 7 m s$^{-1}$. The back trajectory analysis suggests that the high concentrations during this period were most likely associated with the long-range transport of aerosol particles from the western coast of the Iberian Peninsula. In contrast to the summer period, $PM_{10}$ mass concentrations were lower during the fall, despite some temporal peaks. The $PM_{10}$ concentrations were generally lower in September (24.2 ± 5.1 µg m$^{-3}$) and October (30.5 ± 10.8 µg m$^{-3}$). During this period, winds originated from the northeast suggesting the influence of air mass transport from the Mediterranean Sea coast. A sharp fall in PM concentrations was noticed in November (22.8 ± 7.9 µg m$^{-3}$) and December (15.9 ± 5.6 µg m$^{-3}$). Overall, $PM_{10}$ concentration decreased from the summer to winter by 32%. This trend is most likely due to the increased amount of precipitation (peaks of 852mm) during fall and winter, which can lead to the wash-out effect of aerosol and its components (Holst et al., 2008)."*

Figure 7 has been updated according to the new air mass classification. The category of Background Air Masses has been added as follow:

[Figure]

**Figure 7.** Crustal enrichment factors (EF) of aerosol $PM_{10}$ evaluated for the different trace metal elements at AM5; The averaged values are plotted according to their respective air mass origins.

The correlation plot displayed in Figure 8 have been updated according to the new air mass classification as follow:

[Figure]

**Figure 8.** Scatter plot of (A) $NH_4^+$ with $NO_3^-$ and nss-$SO_4^{2-}$ during summer; (B) $NH_4^+$ with $NO_3^-$ and nss-$SO_4^{2-}$ during autumn-winter; (C) $Na^+$ and $Cl^-$ + $SO_4^{2-}$ during ACE air mass; (D) $Na^+$ and $Cl^-$ + $NO_3^-$ during MCE air mass at the AM5 site.

The inter-relationship between aerosol components section (3.4) has been restructured by adding subheadings as follow:

*3.4.1 Nitrate and nss-sulfate*

*3.4.2 Ammonium nitrate and ammonium sulfate*

*3.4.3 Sodium and chlorine*

Therefore, the discussion paragraphs of different correlations have been reformulated as follow:

[revised manuscript text omitted]

**R1-C17:** Line 260: Either delete "while" or replace it with "However" or combine this sentence with the previous sentence - separating the two as "mineral dust, while...".

**R1-A17:** "While" was delated and replaced by "However" in Lines 262-263, and now reads:

*"However, method 2 allows to take into account the overall mineral composition and therefore was applied in this study."*

**R1-C18:** Line 293: replace "accordingly" with "for example,"

R1-A18: The word "accordingly" was replaced with "for example,"

The corresponding sentence has been modified on lines 298-300, which now reads:

*"During August, the high $PM_{10}$ concentrations were mostly related to high wind speeds from the southeast. For example, $PM_{10}$ mass concentration often exceeded 50 $\mu g$ $m^{-3}$ and sometimes even reached up to 145 $\mu g$ $m^{-3}$ during August, when the wind speed was stronger than 9 m $s^{-1}$."*

**R1-C19:** Line 294: says $PM_{10}$ peaked at 143 $\mu g/m^3$ but line 307 says the peak was 145 $\mu g/m^3$.

**R1-A19:** The corresponding $PM_{10}$ concentration has been corrected. The corresponding sentence has been modified on lines 289-290, which now reads:

*"The $PM_{10}$ mass concentration time series at the AM5 station varied from 9.5 $\mu g$ $m^{-3}$ to 145.6 $\mu g$ $m^{-3}$ with averaged of 29.2 ± 17.3 $\mu g$ $m^{-3}$."*

**R1-C20:** Line 336: delete "In contrast" (and in any case, probably don't compare AM5 to a complex urban site outside Morocco as mentioned earlier.)

**R1-A20:** The comparison with other urban sites has been removed and replaced by urban Moroccan sites. In addition, the term „In contrast" was deleted.

***R1-C21:*** Lines 348-349: I assume they mean the urban sites in Morocco are twice or thrice (not trice; also, "two or three times" is better) the AM5 values, but the sentence is unclear.

***R1-A21:*** Indeed, a typing error was made in the sentence. Therefore the sentence has been rephrased in lines 340-343, and now reads:

*"Other Moroccan sites, such as Marrakech, Meknes, and Agadir, which are exposed to strong urban emissions, usually show $PM_{10}$ concentrations between 50 and 110 $\mu g\,m^{-3}$, which is much higher than the concentration found at AM5 in this study (Inchaouh et al., 2017; Tahri et al., 2012; Tahri et al., 2017)."*

***R1-C22:*** The first two sentences of this paragraph could be easily rephrased as "Urban sites show PM10 values two to three times that observed at AM5 (Table 3)." However, I just noticed only one other site in Morocco listed in Table 3 - Tetouan, which has very similar $PM_{10}$ to AM5.

***R1-A22:*** In addition to the Tetouan site, other Moroccan sites such as Marrakech, Kenitra, and Meknes, have now been added in Table 3 as presented previously. Therefore, the sentences in lines 340-345 have been revised and now read:

*"Other Moroccan sites, such as Marrakech, Meknes, and Agadir, which are exposed to strong urban emissions, usually show PM10 concentrations between 50 and 110 $\mu g\,m^{-3}$, which is much higher than the concentration found at AM5 in this study (Inchaouh et al., 2017; Tahri et al., 2012; Tahri et al., 2017). These results highlight a better air quality at AM5 in comparison to many sites and indicate that the station can serve as a good remote reference station for defining background concentrations in Morocco and possibly the whole of North Africa."*


**R2-C1:** This paper presents results from 12-hour PM10 filter samples taken on a newly established high-altitude site (AM5) in Morocco in the middle Atlas. The filters were analyzed for inorganic and organic constituents of particulate matter. The analysis takes into account the meteorological situation like backward trajectories and wind speed. Such data sets are very important for the scientific community, because information on aerosol composition at such remote sites are sparse. The technical aspects of filter analysis are not my field of expertise, but I believe that such a well know institute as TROPOS knows very well how to conduct filter analysis properly. Overall I think the manuscript deserves publication, although I have some concerns about the further data analysis (after the chemical analysis of the filters) and presentation of the results. Since my suggestions include re-structuring of the manuscript, I list them as "major revisions":

**R2-A1:** The manuscript has been copy-edited and large parts of the manuscript have been thoroughly improved. Figures have been improved. Grammatic errors have been corrected and the language has been edited to be more concise throughout. The references have been edited to avoid errors in the citations. The structure of the manuscript has been improved. The classification criteria have been reconsidered, therefore some of the results have been modified. Some sections have been combined to improve readability, comparisons, and avoid repetition.

A point-by-point description of the changes made to the manuscript is listed in the following.

**R2-C2:** I suggest to re-structure the results section as follows: Start with 3.1 (Variation of PM10 mass), then continue with 3.4. (Characterization of aerosol chemical composition for the complete measurement period). Then 3.3 (Air mass classification), then 3.7. (day-night), then 3.6 (dust vs non-dust) and only then 3.2 (Aerosol composition under remote background conditions). Thus, I mean ordering the sections from "robust classification" down to "less well-defined" classification.

**R2-A2:** The results section has been restructured according to the following proposed structure. However, the classification of air masses (3.3) was interchanged with the characterization of Aerosol chemical composition of the complete measurement period (3.4) to facilitate the referencing to the air mass history. Furthermore, as also suggested by the other reviewer (R1-C12), the background conditions have been considered as a separate air mass category. Consequently, its discussion has been incorporated into that of the updated air mass classification (3.3) and the characterization of the aerosol chemical composition (3.4). The chemical composition corresponding to each air mass has been updated according to the new classification which leads to a change in the figures and tables in section (3.3). This restructuring facilitates its comparison with the other categories and also avoids repetition. The new structure is now:

*3.1 Variation of PM10 mass*

*3.2 Air mass origins*

*3.3 Characterization of aerosol chemical composition for the complete measurement period*

*3.4 Inter-relationship between aerosol components*

*3.5 Day and night-time variation*

*3.6 Differences in chemical composition between dust and non-dust events*

**R2-C3:** Furthermore, I suggest to make the same type of plots (bar graphs or pie charts) for all these classifications. It makes comparisons between different separation criteria much easier.

**R2-A3:** We thank the reviewer for the suggestion. The same type of plot, i.e. bar graphs has now been adopted for all the comparisons.

**R2-C4:** Please give more details on trajectory treatment. Line 456/457 says "The air mass origins were classified into four major categories, as showed in Fig.5.". But how exactly was that done?

**R2-A4:** The trajectory analysis was done by grouping samples with similar 12 h trajectories into a cluster and attributing them to a given air mass. If 60% of the hourly trajectories were similar, the sample was allocated to a given air mass category. If they were below this threshold they were considered as mixed and not considered in the classification. In the revised manuscript, corresponding lines have been added on lines 278 to 282 which now read:

*"To group the back trajectories into distinct transport patterns, a manual classification approach was used. This method consists of grouping 12 trajectories with the time interval between adjacent nodes of 1 h and calculated for 96 h, and attributing them to a specific air mass category. The assignment of the trajectories was based on their crossing over given latitude-longitude grids attributed to given geographical sectors. To assign a sample to a specific air mass category, 60% of the trajectories must have a similar profile."*

**R2-C5:** Did you check whether a trajectory crossed a certain latitude-longitude range?

R2-A5: Yes, the trajectories are grouped according to their crossing over given grids, which consist of different longitudes and latitudes. These longitudes and latitudes were attributed to geographical sectors for the better identification of the air mass trajectory. The air masses were then classified considering their geographical origin and four major sectors as follow:

NW sector: Air mass coming from the coast of Europe and crossing the North of Morocco.

W sector: Air mass coming from North Atlantic.

NE sector: Air mass coming from the Mediterranean.

SW sector: Air coming from Saharan dust located at the south-west

**R2-C6:** How many points were used to assign a trajectory to a certain category?

**R2-A6:** Each sample contained 12 hourly 96 h back trajectories with a 1h time interval between adjacent nodes. When 60% of the trajectories had a similar profile, they were assigned to a given category. Scenarios representing mixed cases were excluded in the classification as explained above.

Two corresponding sentences have been added on lines 280-282, which now read:

*"The assignment of the trajectories was based on their crossing over given latitude-longitude grids attributed to given geographical sectors. To assign a sample to a specific air mass category, 60% of the trajectories must have a similar profile."*

**R2-C7:** Did you use a clustering algorithm?

**R2-A7:** We did not apply a clustering algorithm; we did a visual classification based on the frequency of trajectories with similar profiles. However, the obtained results were compared with those of a cluster analysis approach according to Cui et al. (2021) using the HYSPLIT algorithm (Rolph et al. 2017). The results of the air mass categories of the cluster analysis were identical to those obtained with the above approach but differed by about 8% in terms of the frequency associated with the categories above. The results of the comparison have been presented as Supplementary information as Figure S1.

[Figure]

**Figure S1.** Cluster analysis of back trajectories arriving at AM5 site from August to December 2017 classified into 4 trajectory clusters

**R2-C8:** Did you consider the vertical motion? It makes a big difference if a trajectory crosses the desert at 2000 m or at 200 m altitude.

**R2-A8:** The vertical motion was taken into consideration during the backward trajectory calculations and the classification process. The trajectories have been calculated for three different heights (100 m, 500 m, 1000 m) above ground level through the HYSPLIT model. The vertical transport was modeled using the isobaric option of HYSPLIT.

**R2-C9:** What exactly is shown in Figure 5?

**R2-A9:** Figure 5 presents 96-hour HYSPLIT backward trajectories shown separately for typical samples that represent an air mass category. The back trajectories were imported as GIS files and plotted using the QGIS software program. The dates for each sample as well as their $PM_{10}$ mass has been added in the caption of the Figure. The Figure is now updated to include a typical background condition air mass. The new Figure and caption are presented below.

[Figure]

**Figure 4.** Typical 96h air mass back trajectory performed for AM5 during routine samples periods; aerosol type and $PM_{10}$ mass concentration are given in parentheses: (a) 18 December 2017: air mass from the North Atlantic Ocean considered to be representative of background conditions (Background, $m$=10.9 µg m$^{-3}$); (b) 10 October 2017: air mass from Europe crossing coastline of North Morocco (Atlantic Coast Europe, $m$=44.1 µg m$^{-3}$); (c) 2 November 2017: slightly polluted air mass from North East crossing Mediterranean Sea Morocco (Mediterranean Coast Europe, $m$=26.4 µg m$^{-3}$) ; (d) 13 August 2017: dust loaded air mass coming from Sahara desert (Saharan Dust, $m$=94.1 µg m$^{-3}$).

The paragraph describing the results of updated air mass classification in section air mass origins (3.2) has been modified. The corresponding sentences have been modified in lines 347-357 and now read as follows:

*"The calculation of back trajectories using the HYSPLIT model allowed the identification of several remote sources of PM10 at the station. Four main air masses categories were identified as shown in Fig. 4. i) Air masses that spent the last 96h over the Atlantic Ocean at high-altitude (1000 m.asl), representative of typical background air mass (BAM) conditions, which influenced about 5.3% of all samples; ii) Air masses originating from the Atlantic and crossing over the Coast of Europe (ACE), especially Spain and over Moroccan industrial cities located at the North Atlantic coast, influencing about accounts 26.8% of all samples; iii) Air masses from Europe crossing over the Mediterranean Coast and Europe (MCE) as well as over North Moroccan cities, as shown in Fig. 4c and influencing about 37.4% of all samples and iv) Air masses originating from Southern and/or Eastern Sahara crossing the desert (SD), at different altitudes before arriving at the AM5 and influencing about 22.6% of all samples. The remaining back trajectories represent mixing scenarios including 7.9% of all samples that could not be assigned to the four major classes mentioned above."*

**R2-C10:** How many trajectories of sample times are not shown here?

**R2-A10:** Some of the samples which represent 8% didn't match the criterion explained above and were representative of the mixed scenario and were, therefore, excluded (n=15). An example is shown below:

A corresponding line has been added on the manuscript on lines 282-283:

*"Samples (n=15 samples) with trajectories from mixed origins (e.g., marine air mass over Europe and the desert) were excluded from the classification of the air masses."*

**R2-C11:** The caption says "PM10 mass concentration are given in parentheses" but only a percentage value is given.

**R2-A11:** The date and the mass concentration of each distinctive sample have been added to the figure caption as presented above.

**R2-C12:** Did you check how the local orography is represented in the model? 2000 m starting point may be too high if the model landscape smoothes the actual mountain range down to lower altitudes. HYSPLIT offers the option "above ground level", so initializing the trajectories with "10 m above ground level" might be a good sensitivity test.

**R2-A12:** We thank the reviewer for pointing out the choice of arrival altitudes for the back trajectory calculation, which is an important step. There was an error in the manuscript, the sentence referred to 2000m above sea level and not ground level. However, as advised, a sensibility test was performed using trajectory calculations at 2000 m above sea level and 10 m, 100m, and 1000m above ground level as references. The trajectories at 1000 m above ground level were found stable and consistent with the trajectories at 2000 m above sea level since the background level height (terrain height) of the grid cell used by HYSPLIT and the GDAS 1 files was 1000 m in this location. The Figure below presents the results of the comparison between 2000m a.s.l and 1000 m a.g.l. The comparison shows similar trajectories. It proves that the sensitivity of the land level is more relevant. Therefore, as suggested by the reviewer, we have recalculated the trajectories using the ground level with an altitude of 1000 m above ground level as a reference as shown in Fig.1 of the supplement. A corresponding line has been added on the manuscript on lines 275-276:

*"Due to the resolution of the input data, the exact altitude of the mountain is not properly represented and according to the HYSPLIT model at the AM5 site, the terrain height is at 1000 m only. Therefore, trajectories were calculated every hour for the altitude of 1000 m above model ground."*

[Figure]

**Figure S2.** Back trajectory calculated at 1267 m above ground level and 2000 m above sea level.

**R2-C13:** The definition of the background conditions ("mass concentration lower than 20 µg/m3 and low wind speeds less than 4 m/s.") is not clear to me and seems artificial.

**R2-A13:** This issue was also risen by R1 (comment R1-C4). The choice of the background definition was motivated by different studies such as Puxbaum et al., 2004; Vardoulakis and Kassomenos, 2008; Karaca et al., 2009; Harrison et al., 2004; Escudero et al., 2006 that suggests that the reference could be at 20 µg m$^{-3}$. However, as recommended by R1, the lowest fifth (5$^{th}$) percentile of PM$_{10}$ mass concentrations has now been considered as the baseline threshold. This can be easily seen in the new PM$_{10}$ variation figure (Fig. 3A). Based on these new criteria, the PM$_{10}$ < 12 µg/m$^3$ was considered as background. The number of samples that med these conditions was 10 and the mean concentration was about 10.9 µg m$^{-3}$.

A corresponding text has been added on the manuscript on lines 318-329:

*"To establish a reference baseline and evaluate the background conditions at the site, the lower 5$^{th}$ percentile of the PM$_{10}$ concentrations was found to be representative of remote background aerosol conditions. The PM$_{10}$ frequency and probability density function as shown in Fig. S3 confirmed this observation. The samples within this PM concentration range were had similar air mass trajectories and typical meteorological conditions with low wind speeds < 3 m s$^{-1}$. The air masses typically traveled in the free troposphere at about 1000 m above sea level, crossing the North-Atlantic Ocean before arriving at the site within the past 96 h. These conditions were, however, not free from local and regional pollution from point sources such as dust resuspension from the cars assessing the site.*

*Consequently, the average background PM$_{10}$ mass concentration at the Middle-Atlas was 10.9 µg m$^{-3}$, which was found to be stable and representative of periods of little external influence. In comparison, Benchrif et al. (2018) reported background PM$_{10}$ values for Northern Morocco with an average of 12.2 µg m$^{-3}$, which is very similar to the concentrations determined in this study, 10.9 µg m$^{-3}$."*

[Figure]

**Figure 3.** (A) Time series of daily PM$_{10}$ mass.

**R2-C14:** More data analysis is required for such a classification. For example, create a PDF (probability density function) of mass concentrations and wind speeds to show that the chosen thresholds are reasonable.

**R2-A14:** We thank the reviewer for this suggestion. Accordingly and also a reply to R1-C2 the frequency and the probability density function (PDF) of mass concentrations and wind speed were calculated for background samples. The thresholds chosen are in line with the results obtained from statistical calculations, as highlighted with the blue bars in the frequency and probability density function Figures below.

[Figure]

**Figure. S3** PM$_{10}$ mass and wind speed probability density functions during the sampling period.

We have chosen to include these Figures into the SI. A corresponding sentence has been added to the manuscript on line 320:

*"The PM$_{10}$ frequency and probability density function as shown in Fig. S3 confirmed this observation."*

**R2-C15:** Trajectory information may also be used (see comments above) so check whether the air masses touched the boundary layer or travelled only in the free troposphere.

**R2-A15:** Trajectory information was used to select the background samples. The samples were of air masses that traveled in the free troposphere at altitudes 1000 m above sea level coming from North-Atlantic Ocean within the past 96 h before arriving at the site. This aspect has been further highlighted in the revised version under section 2.5, lines 322-323:

*"The air masses typically traveled in the free troposphere at about 1000 m above sea level, crossing the North-Atlantic Ocean before arriving at the site within the past 96 h."*

**R2-C16**: A classification based on solid, statistical relevant criteria is needed for this section.

**R2-A16:** The definition of the background and the selection criteria have been reviewed and are given as follows:

- Samples that lie within the lower $5^{th}$ percentile of $PM_{10}$ mass concentrations

- Probability density function ($PM_{10}$) indicating a range of the lower $5^{th}$ percentile

- The Wind speeds lower than 3 m s$^{-1}$

- Samples were typical of air masses coming from the North-Atlantic Ocean crossing free troposphere at altitudes above 1000 m during transport

**Further comments**

**R2-C17:** Section "3.1.2 comparison with other stations" and (Table 3):I miss Jungfaujoch and other GAW stations. There is a data base of the GAW stations: https://gawsis.meteoswiss.ch/GAWSIS/#/

**R2-A17:** Urban stations especially in India or China have been removed from the table and replaced by remote high-altitude stations and urban measurements made in Morocco. The station Jungfraujoch and other GAW stations such as Mt. Everest, Mt. Cimone, and puy de Dôme were added to Table 3 to make the comparison more relevant. Other data from urban sites located in Morocco such as Marrakech, Meknes, and Kenitra have also been added. Therefore, Table 3 has been updated and the discussion of the comparison with other sites in section 3.1 on the manuscript has been rewritten and now read in lines 231-345:

[revised manuscript text omitted]

*In lines 517-519: "The concentrations of sulfate (0.9 ± 0.8 $\mu g \ m^{-3}$) over AM5 were comparable with those at Puy de Dôme (1.3 ± 1.1 $\mu g \ m^{-3}$), but were almost 4-5 times lower than all the other hilly stations except Mt. Everest (Decesari et al., 2010)."*

**R2-C18:** Figure 6: Background shading type is too coarse, hard to recognize

**R2-C18:** The background shading type has been modified to improve visibility. Thus, each sample is represented by a distinguishing color symbol for each specific air mass. At the same time, Figure 6, which shows the temporal variation of the organics, has been modified in the same way. The new Figure 5 and Figure 6 are shown below:

[Figure]

**Figure 5.** Time series of major aerosol chemical constituents in $PM_{10}$ filter samples collected from August to December 2017; The color of the symbols displayed for each sample represents a specific air mass origin: Background (blue); ACE (red); MCE (green); SD (yellow).

The description of Figure 6 has been modified in section 3.3 according to the new air mass classification, so the modified paraphrases are listed as follows:

In lines 364-368:" During the 5 months of PM collection at the AM5 site, the average mineral dust concentration was about $17.7 \pm 7.4$ $\mu g$ $m^{-3}$ and varied strongly between 0.05 $\mu g$ $m^{-3}$ and 107 $\mu g$ $m^{-3}$. The highest mean concentrations were observed in August (39 $\mu g$ $m^{-3}$) and the lowest in December (3.7 $\mu g$ $m^{-3}$). Low concentrations were observed during days with low wind speeds (< 2 $m$ $s^{-1}$), low Saharan dust air mass inflow, and after precipitation events, which typically occurred in the fall and winter."

[revised manuscript text omitted]

Figure 7 has been updated according to the new air mass classification. The category of Background Air Mass has been added as follow:

[Figure]

**Figure 7.** Crustal enrichment factors (EF) of aerosol $PM_{10}$ evaluated for the different trace metal elements at AM5; The averaged values are plotted according to their respective air mass origins.

The correlation plot displayed in Figure 8 have been updated according to the new air mass classification as follow:

[Figure]

**Figure 8.** Scatter plot of (A) $NH_4^+$ with $NO_3^-$ and nss-$SO_4^{2-}$ during summer; (B) $NH_4^+$ with $NO_3^-$ and nss-$SO_4^{2-}$ during autumn-winter; (C) $Na^+$ and $Cl^- + SO_4^{2-}$ during ACE air mass; (D) $Na^+$ and $Cl^- + NO_3^-$ during MCE air mass at the AM5 site.

The inter-relationship between aerosol components section (3.4) has been restructured by adding subheadings as follow:

*3.4.1 Nitrate and nss-sulfate*

*3.4.2 Ammonium nitrate and ammonium sulfate*

*3.4.3 Sodium and chlorine*

Therefore, the discussion paragraphs of different correlations have been reformulated as follow:

[revised manuscript text omitted]

**R2-C19:** Data availability: Please make the data publicly available through a database or repository. The data set should be easy to handle (a simple table with date, time, concentration of compounds) for everybody.

**R2-C19:** The data are available upon request and will be uploaded also on the Pangaea database.

---

## Author Response (AR2)

We thank the reviewer for the many constructive comments and suggestions that have enable us to improve on the manuscript. Comments by the reviewer are given in black, our response to the comments are shown in red, and modified text in the revised manuscript are given in green. In the following, we have addressed these concerns in a point-by-point format.

**Anonymous Referee #2, 05 October 2021**

The authors have greatly improved the manuscript based on the reviewers' suggestions, so I recommend publication now.

I found a few typos and small mistakes:

The Spelling errors and grammatic errors have been corrected throughout the manuscript and the language has been edited to be more concise throughout. In addition, the name of the Atlas Mohammed V atmospheric observatory was replaced from "AM5" to "AMV".

A point-by-point description of the changes made to the manuscript is given in the following:

Line 24:"ranged" -> "range"

Ranged has been changed to range. Line 24 now reads:

*"During background conditions characterized by low wind speeds (Av. 3 m s⁻¹) and mass concentrations in the range from 9.8 to 12 μg m⁻³."*

Line 373: "...the...mountains ... act..."

Acts has been changed to act. Line 373 now reads:

*"Thirdly, the High-Atlas Mountains situated at 4000 m of altitude act as a barrier to Saharan dust transport which forces the winds to deviate from their path."*

Line 711 "vanadium"

"Vanadium" has been changed to "vanadium". Line 711now reads:

*"The nss-sulfate concentrations were slightly correlating with vanadium which is associated with the emissions of oil combustion, ship emissions as well as iron and steel industrial emissions (Pandolfi et al., 2011)."*

Line 823 "The long-range..."

"Long-range" has been changed to "long-range". Line 823 now reads:

*"The long-range transport of mineral dust showed a significant impact on $PM_{10}$ composition."*

Line 850: "Sulfate, nitrate, calcium, and ammonium...

"Sulfate" has been changed to "sulfate". Line 850 now reads:

*"Sulfate, nitrate, calcium, ammonium, showed an increase in the average concentration of about a factor of 4.5 while chloride, magnesium, and potassium experienced an increase in their concentrations by a factor of 5 (Fig 10c, 10d)."*

Fig. 6: please check Y-axis labels (µg or ng)

The unit of n-alkanes and PAHs was corrected in Fig. 6 from *"µg/m³"* to *"ng/m³"*. The modified Figure 6 are shown below:

[Figure]

**Figure 6.** Time series of organic compounds in PM$_{10}$ filter samples collected from August to December 2017 at AM5; The color of the symbols displayed for each sample represents a specific air mass origin: Background (blue); ACE (red); MCE (green); SD (yellow).

Other spelling errors and grammatic errors have been corrected throughout the manuscript as follow:

"AM5" has been changed to "present work". Lines 15-16 now read:

*"The main objectives of the present work are to investigate the variations in the aerosol composition, and better assess global and regional changes in atmospheric composition in North Africa."*

"Assess" has been changed to "to assess". Lines 103-104 now read:

*"The evaluation of local regional, trans-regional, and climate change effects can help to assess air quality and climate-relevant mitigation strategies."*

"In the valleys during high-pressure weather" has been changed to "during high-pressure weather in the valleys". Line 144 now reads:

*"Thermal breezes are common during high-pressure weather in the valleys."*

The Middle-Atlas region is considered "very humid" and "temperate" have been changed to "to be a very humid" and "with temperate", respectively. Line 154 now reads:

*"The Middle-Atlas region is considered to be a very humid and with temperate climate."*

"The limit of detection" has been changed to "detection limit". Line 192 now reads:

*"The detection limit for OC/EC measurement was 0.2 $\mu g/cm^2$."*

"Potassium, calcium, magnesium, and sulfate" has been changed to "Potassium (ss-$K^+$), calcium (ss-$Ca^{2+}$), magnesium (ss-$Mg^{2+}$), and sulfate (ss-$SO_4^{2-}$)". Lines 229-231 now read:

*"Sea salt concentrations were calculated by adding chloride to sodium, and the sea salt (ss) contributions of potassium (ss-$K^+$), calcium (ss-$Ca^{2+}$), magnesium (ss-$Mg^{2+}$), and sulfate (ss-$SO_4^{2-}$), , have been estimated as 0.03, 0.5, 0.12, and 0.25 fractions of the measured $Na^+$, respectively (Marenco et al., 2006)."*

"come" has been changed to "blow". Line 806 now reads:

*"Indeed, the winds blows from all directions, but it is dominated from the east section during the day, and by the west during the night, as shown in Fig. S10."*

"distant" has been changed to "distance". Line 808 now reads:

*"This suggests that long-distance transport often occurred during the night while the wind speed during the day was relatively lower."*

"Nonacosane and Heptacosane" has been changed to "nonacosane and heptacosane". Lines 869-871 now read:

*"During this period, a correlation was observed between OC as well as some organic species nonacosane and heptacosane with mineral elements such as calcium ($r^2=0.81$) and magnesium ($r^2=0.73$)."*